# Enabling Instructional Image Editing with In-Context Generation in Large Scale Diffusion Transformer

**Zechuan Zhang**[1], **Ji Xie**[1], **Yu Lu**[1], **Zongxin Yang**[2], **Yi Yang**[1†]
[1]ReLER, CCAI, Zhejiang University
[2]DBMI, HMS, Harvard University
Project Page: https://river-zhang.github.io/ICEdit-gh-pages

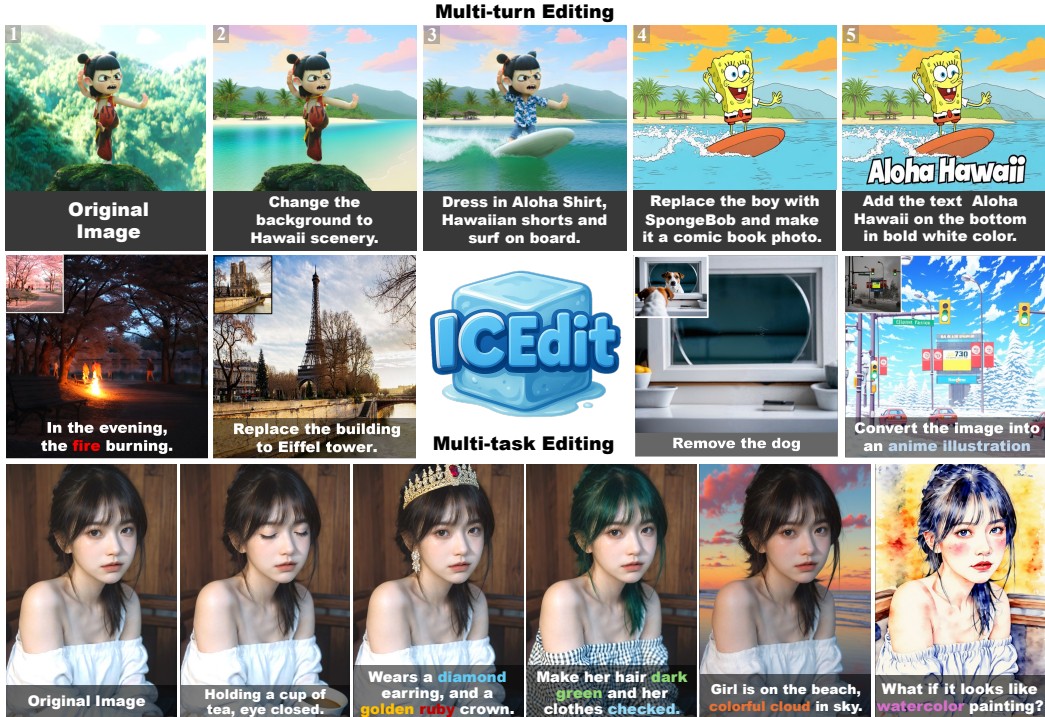

Figure 1: We introduce ICEdit, a novel method that achieves state-of-the-art instruction-based image editing with only 0.1% training data required by previous SOTA methods, demonstrating exceptional generalization. The first row illustrates a series of multi-turn edits, executed with high precision, while the second and third rows highlight diverse, visually impressive editing results from our method.

## Abstract

Instruction-based image editing enables precise modifications via natural language prompts, but existing methods face a precision-efficiency tradeoff: fine-tuning demands massive datasets (>10M) and computational resources, while training-free approaches suffer from weak instruction comprehension. We address this by proposing **ICEdit**, which leverages the inherent comprehension and generation abilities of large-scale Diffusion Transformers (DiTs) through three key innovations: (1) An in-context editing paradigm without architectural modifications; (2) Minimal parameter-efficient fine-tuning for quality improvement; (3) Early Filter Inference-

39th Conference on Neural Information Processing Systems (NeurIPS 2025).

Time Scaling, which uses VLMs to select high-quality noise samples for efficiency. Experiments show that ICEdit achieves state-of-the-art editing performance with only 0.1% of the training data and 1% trainable parameters compared to previous methods. Our approach establishes a new paradigm for balancing precision and efficiency in instructional image editing.

# 1 Introduction

In recent years, instruction-based image editing has gained considerable attention for its ability to transform and manipulate images using natural language prompts. The main advantage of instruction-based editing is its ability to generate precise modifications with minimal textual instructions, thereby opening new possibilities for both automated image processing and user-driven content creation.

Instruction-based image editing methods are divided into finetuning-based [1, 2, 3, 4, 5, 6, 7, 8, 9, 10] and training-free approaches [11, 12, 13, 14, 15, 16, 17, 18, 19]. Finetuning-based methods achieve precise instruction-following by fully finetuning pretrained diffusion models on large-scale datasets (450K to 10M samples [1, 3]) with structural modifications like condition embedding [9, 5] or channel adjustments [1, 2, 4], but demand significant computational resources. In contrast, training-free methods avoid retraining through techniques like image inversion, or attention manipulation, offering efficiency but struggling with complex instructions, which reduces precision and practical utility. This highlights **a critical trade-off between precision and efficiency** in current methods.

Despite the dilemma above, recent advances in diffusion transformers (DiT) [20, 21, 22] suggest a promising pathway. DiT architectures exhibit two critical properties: (1) **Scalable Generation Fidelity**: Larger DiT variants (e.g., FLUX [23]), trained on vast amounts of image-text data, possess unprecedented text-to-image alignment capabilities. (2) **Intrinsic Contextual Awareness.** Diffusion Transformers (DiTs) leverage attention mechanisms to enable bidirectional interactions between reference and generated content, processing source and target images concurrently. This facilitates tasks like reference-guided synthesis [24, 25] and identity-preserved editing [26], while supporting conditional image generation without specialized alignment networks [27, 7, 28].

Although these works achieve promising results, they are unsuitable for instruction-guided image editing due to their limited ability to comprehend explicit editing instructions and preserve the layout of non-editable regions. This raises a critical question: **Can large scale DiT's generation capacity and contextual awareness directly address instruction-based image editing**, while **balancing precision and efficiency through intrinsic capabilities rather than external complexity?**

Our experiments reveal two fundamental limitations in using DiTs for instructional image editing: (1) **Poor instruction comprehension**: while the model can interpret descriptive input/target prompts, it struggles with direct editing instructions (e.g., "make it..." or "change it..."); (2) **Layout instability**: the model often alters unchanged regions when regenerating scenes, leading to poor editing success.

To address these limitations, we suggest a two-part solution. First, we recommend turning direct editing commands into descriptive prompts that match how DiTs naturally understand information. Second, the editing challenges mainly arise from the model's unlearned image-to-image editing priors, which can potentially be addressed through lightweight fine-tuning with editing pairs or test-time adaptation [29, 30] strategies. This approach offers the potential to simultaneously resolve DiTs' two fundamental limitations in instructional editing while maintaining precision-efficiency balance.

In this paper, we propose **ICEdit, an efficient and effective framework** for instructional image editing that directly exploits the inherent comprehension and generative capabilities of large-scale DiT priors for instructional image editing. Our approach employs in-context prompts—a fixed-format prefix embedding editing instructions in a format DiT can effectively interpret (§3.1). Given a source image (left panel), the model generates edited outputs (right panel) by jointly processing these prompts and the input (see fig. 2(a) and fig. 5). Moreover, minimal parameter-efficient fine-tuning significantly improves editing success and quality, while adopting a Mixture-of-Experts (MoE) [31] further enhances performance (§3.2). Finally, we introduce *Early Filter Inference-Time Scaling*, which uses vision-language models (VLMs) to evaluate noise quality during early denoising stages in rectified flow models (§3.3). This method rapidly identifies and filters out noises congruent with textual instructions, enhancing robustness and output fidelity.

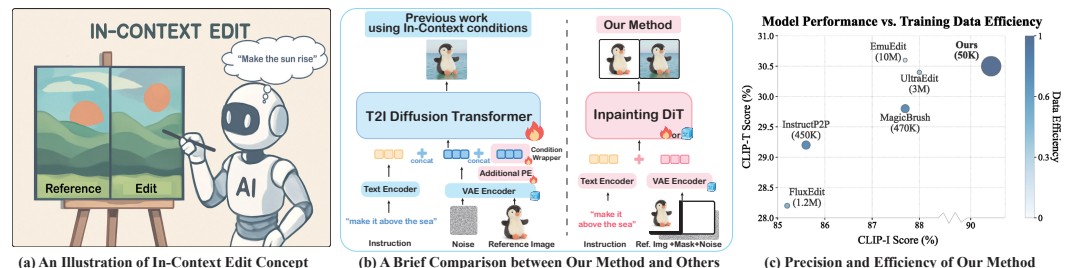

| (a) An Illustration of In-Context Edit Concept | (b) A Brief Comparison between Our Method and Others | (c) Precision and Efficiency of Our Method |

Figure 2: (a) Our concept: The original DiT model is conceptualized as a painter who generates edited images (right panel) by interpreting the reference image (left panel) and the instruction, similar to finishing an artwork based on a provided sketch. (b) Comparison of model structures: Unlike prior DiT methods, our approach avoids extra position/condition encoders, easily and effectively preserving the original model structure. (c) Our method leverages minimal training data and achieves comparable performance with SOTA models.

We evaluate our method on the Emu Edit [3] and MagicBrush [2] benchmarks, demonstrating three key advancements: **Significant Data Efficiency and Editing Quality**: Achieving state-of-the-art results with minimal training data (0.1% of prior requirements). **Validation of Proposed Paradigm**: Outperforming recent DiT-based editing models (both T2I and inpainting variants), confirming the effectiveness of our in-context editing approach. **Practical Applicability**: Attaining a competitive VIE score of **78.2** (compared to SeedEdit's 75.7), demonstrating real-world viability comparable to commercial systems. These results establish a novel perspective on balancing precision and efficiency (fig. 2(c)) by leveraging large-scale DiTs as priors. Our contributions include:

- We explore the editing ability of large pretrained DiTs and propose an in-context editing paradigm, ICEdit, that enables instructive image editing by leveraging the model's inherent understanding and generation abilities, without requiring architectural modifications or extensive fine-tuning.

- Our framework demonstrates significant improvements through minimal fine-tuning, significantly enhancing editing quality and robustness. We further propose an *Early Filter Inference-Time Scaling* approach that uses VLM to select high-quality noise samples. This integrated strategy improves editing precision while maintaining computational efficiency.

- Experimental results show that our method **achieves state-of-the-art editing performance while requiring only 0.1% of the training data compared to previous approaches**, establishing a novel perspective on balancing precision and efficiency.

## 2 Related Work

**Training-free editing techniques**. Since the emergence of Diffusion Models, numerous training-free image editing methods [11, 32, 19, 33, 15, 17, 16, 13] have gained attention. Recently, RF-Solver [13] improves inversion precision in Rectified-flow models by mitigating ODE-solving errors and leverages MasaCtrl [19] for image editing. StableFlow [14] identifies critical MM-DiT Blocks through ablation studies, injecting features only into these blocks to enhance editing capabilities. However, these methods face two key limitations: 1) manually designed modules restrict generation ability, hindering complex instruction understanding and reducing success rates; 2) editing requires carefully crafted prompts, limiting generalizability and scalability.

**Finetuning-based editing methods**. Most current editing models modify architectures and fine-tune on high-quality datasets [1, 2, 34, 4, 35, 10, 8, 36, 37, 38, 39, 40, 41]. InstructPix2Pix [1], MagicBrush [2], and UltraEdit [4] fine-tune diffusion UNet using original images as input. MGIE [5] and SmartEdit [9] enhance instruction understanding by integrating a Multimodal Large Language Model (MLLM) to encode and inject instructions into the diffusion model. However, **a gap exists between the embedding spaces of generative prompts and editing instructions**, reducing the generalization ability of Diffusion Models and necessitating large datasets to bridge it. For instance, InstructPix2Pix generated 450K pairs, Emu Edit [3] collected nearly 10M pairs, and FluxEdit [42] used 1.2M pairs from [34] based on FLUX [23], yet the editing results remain suboptimal.

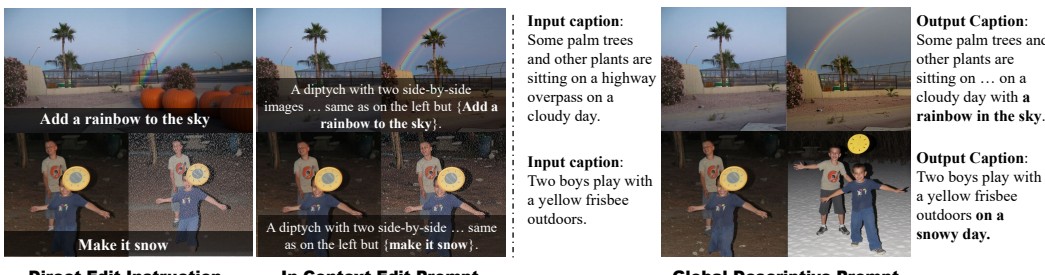

**Direct Edit Instruction**  **In-Context Edit Prompt**  **Global Descriptive Prompt**

Figure 3: **Input Prompt Variants for Image Editing**. We evaluate three prompt formats: (1) Direct Instruction - explicit editing commands provided directly; (2) In-context Prompt - instructions embedded in structure "A diptych with... On the right, the same scene but {instruction}"; (3) Global Descriptive Prompt - uses full input/output captions ("On the left {input} On the right {output}").

**T2I In-context Editing Framework Results (based on Flux.1 dev)**

**Inpainting In-context Editing Framework Results (based on Flux.1 Fill)**

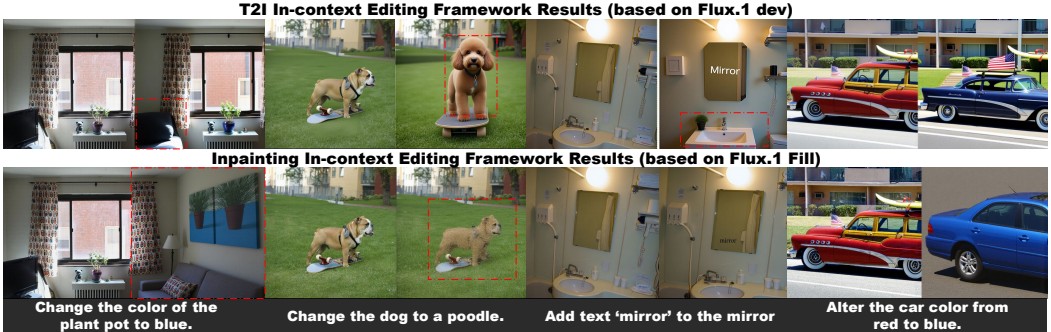

**Change the color of the plant pot to blue.**  **Change the dog to a poodle.**  **Add text 'mirror' to the mirror**  **Alter the car color from red to blue.**

Figure 4: **Training-Free Methods Show Limited Performance.** Both T2I and inpainting DiT frameworks (fig. 5) yield suboptimal results. The T2I DiT struggles to preserve the original layout, while the inpainting framework may inadvertently perform outpainting. Despite these shortcomings, both demonstrate potential in following instructions and modifying edited regions.

## 3 Method

In this section, we first explore in-context editing capabilities within original DiT generative models and propose our **in-context edit framework** for instruction-based image editing (§3.1). After thorough analysis, we perform minimal fine-tuning to significantly improve editing quality and robustness of our editing paradigm (§3.2). Finally, we present **early filter inference-time scaling** (§3.3) to optimize initialization noise for better generation quality.

### 3.1 Exploration of DiT's In-context Edit Ability

**In-Context Edit Framework**. Inspired by recent advances in large-scale Diffusion Transformer (DiT) models [27, 26, 43, 24], which demonstrate robust contextual capabilities, we investigate image editing via in-context generation. As illustrated in fig. 2(a), our approach mimics an AI painter that generates edited images (right panel) by interpreting a reference image (left panel) and an edit instruction. By leveraging DiT's strong generation fidelity and inherent contextual awareness, we aim to enable direct image editing without requiring module modifications or extensive finetuning.

We propose two training-free frameworks based on text-to-image DiT (ICEdit-T2I) and inpainting DiT (ICEdit-Inpaint), respectively, as shown in fig. 5(a). For the ICEdit-T2I framework, we introduce an implicit reference image injection method. We perform image inversion [44, 13, 14, 19, 18] on the reference image, preserving attention values across layers and steps. These values are injected into tokens representing the left side of a diptych for image reconstruction, while the right side is generated based on the edit instruction within a predefined prompt during in-context generation.

Conversely, the ICEdit-Inpaint framework offers a more straightforward approach. By accepting a reference image, we construct a side-by-side image, where the left-hand side is reconstructed as a reference image, and the right-hand side is the "edited" result. The whole process is guided by a **fixed mask**, which consistently covers half of the diptych, and an edit prompt to produce the edited output.

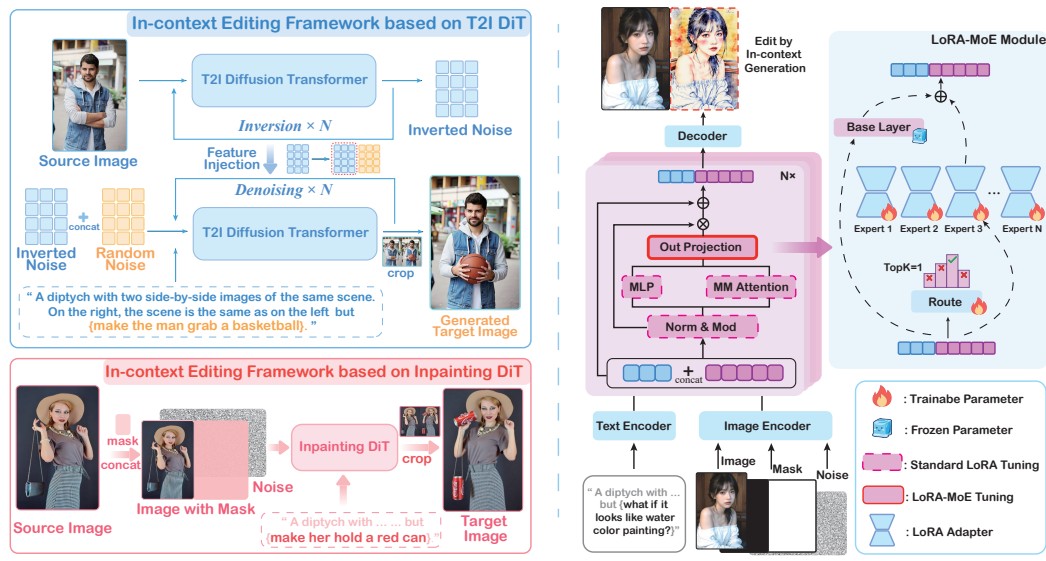

| (a) Training-free Structure of ICEdit | (b) Finetuning Strategy for ICEdit |

Figure 5: **Model Structure.** (a) We propose two training-free architectures for in-context editing using large-scale DiTs, adapted from (a) T2I-DiT and (b) inpainting-DiT. Both adopt a diptych framework: left panel = source image, right panel = editing output, consistent with Fig. 2. (b) While both show limited performance, we adopt the inpainting paradigm for further fine-tuning due to its simplicity (no image inversion required). Our method integrates parameter-efficient adaptation with dynamic expert routing for specialized feature processing.

Unlike prior DiT-based approaches [27, 26, 24], our method eliminates the need for intricate position and condition encoding designs or retraining, as illustrated in fig. 2. Instead, it purely leverages the diptych image structure and the inherent processing capabilities of DiT.

**In-Context Edit Prompt.** Diffusion models typically struggle to interpret editing instructions due to a mismatch between the embedding spaces of descriptive prompts and editing instructions. Previous approaches [1, 2, 3, 4, 42], rely on extensive training with large-scale editing datasets to enable generative diffusion models to understand editing instructions. In contrast, we propose leveraging the inherent contextual understanding of powerful Diffusion Transformers (DiTs) to perform instruction-based image editing without heavy training.

We experimented with three prompt types to enable a DiT model (e.g., FLUX) to edit a given image, as shown in fig. 3. First, directly inputting the editing instruction into the model often fails to produce accurate results, frequently altering the entire image layout. Second, we designed an **in-context edit prompt (IC prompt)** that embeds the instruction within a descriptive structure: "*A diptych with two side-by-side images of the same scene. On the right, the scene is identical to the left but {instruction}.*" This IC prompt significantly improves the model's ability to interpret instructions, yielding approximately a 70% increase in editing success rate, as reported in table 3. Finally, we tested a training-free approach using global descriptive captions for both source and target images, similar to prior methods [44, 14, 16, 13]. While achieving better quality and instruction adherence, this method relies on impractical, detailed image descriptions, undermining seamless instruction-based editing. Thus, we adopt the IC prompt in our framework and further refine it through fine-tuning.

**Discussion of the Training-Free Framework:** Figure 4 presents the editing outcomes of the T2I and inpainting frameworks on the Emu Testset, guided by the IC prompt. While both frameworks demonstrate some editing capability, **their zero-shot performance is unsatisfactory**, particularly in preserving unedited regions, which limits their practical utility (quantitative eval in table 3). We attribute this to the lack of learned image-to-image editing priors. This limitation could be mitigated through lightweight adjustments, such as finetuning or test-time scaling. Given that the ICEdit-T2I framework requires time-consuming image inversion, **we favor the ICEdit-Inpaint framework for its straightforward operation**, which facilitates further finetuning.

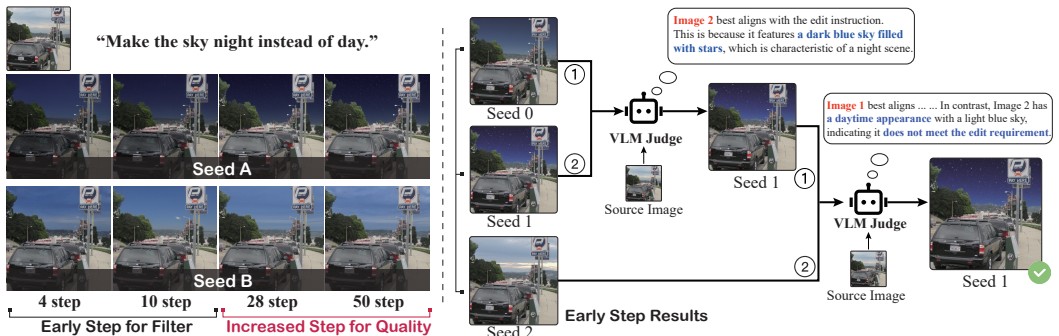

Figure 6: **Illustration of Inference-Time Scaling Strategy (§3.3).** The upper rows demonstrate that edit success can be assessed within a few initial steps. These early results are used to filter the optimal initial noise with VLM judges.

## 3.2 Efficient Fine-tuning for Enhanced Editing

We define our inpainting-based editing method as a function $\mathcal{E}$ mapping a source image $I_s$ and edit instruction $T_e$ to the edited output $I_t$:

$$I_t = \mathcal{E}(I_s, T_e) = \mathcal{D}(I_{IC}, M, T_{IC}), \tag{1}$$

where $\mathcal{D}$ is the inpainting DiT, $I_{IC}$ is the in-context image with $I_s$ on the left, $M$ is a fixed binary mask to reconstruct $I_s$, and $T_{IC}$ is the in-context edit prompt derived from $T_e$.

To boost performance, we curate a compact dataset (50K samples) from public sources (§4) and apply LoRA fine-tuning [45, 27] to multi-modal DiT blocks, achieving a 150% improvement in editing success (table 3) despite the small dataset. However, tasks like style transfer and object removal remain challenging, as a single LoRA structure struggles to handle diverse editing tasks requiring distinct latent feature manipulations.

**Mixture of LoRAs.** To overcome these limitations, we propose a Mixture-of-Experts (MoE) inspired LoRA structure within the DiT block (fig. 5(b)), inspired by MoE architectures [46, 47, 31]. We integrate $N$ parallel LoRA experts into the multi-modal attention block's output projection layer, using standard LoRA elsewhere. A routing classifier selects experts based on visual tokens and text embeddings. Each expert, a LoRA module with rank $r$ and scaling factor $\alpha$, contributes to the output:

$$\text{Output} = \text{BaseLayer}(x) + \frac{\alpha}{r} \sum_{i=1}^{N} G(x)_i \cdot B_i \cdot A_i \cdot x, \tag{2}$$

where $B_i \in \mathbb{R}^{d \times r}$, $A_i \in \mathbb{R}^{r \times k}$, $x \in \mathbb{R}^k$, and $G(x)_i$ is the routing probability. We use a sparse MoE setup, selecting the top-$k$ experts: $G(x)_i = \text{softmax}(\text{TopK}(g(x), k))_i$, where $\text{TopK}(\cdot, k)$ retains the top-$k$ entries, setting others to $-\infty$, ensuring efficiency and versatility for diverse editing tasks.

## 3.3 Early Filter Inference Time Scaling

During inference, we find that initial noise significantly shapes editing outcomes, with some inputs producing results better aligned with human preferences (see fig. 9), a pattern supported by recent studies [30, 29]. This variability drives us to investigate inference-time scaling to improve editing consistency and quality. In instruction-based editing, we observe that **success in instruction alignment often become evident in few inference steps** (see fig. 6), a characteristic compatible with rectified flow DiT models [48, 49]. These models traverse latent space efficiently, delivering high-quality outputs with few denoising steps—sometimes as few as one [50]. Thus, unlike generation tasks that demand more steps for detail and quality, **we can evaluate edit success with only a few steps.**

Based on this insight, we propose an *Early Filter Inference Time Scaling* strategy. We start by sampling $M$ initial noise candidates and generating a preliminary $m$-step edit for each, where $m \ll n$ (the full denoising steps). A visual large language model (VLM) [51, 52, 53] then assesses these $M$ early outputs for instruction compliance, using a bubble sort-inspired pairwise comparison to iteratively pinpoint the top candidate, akin to selecting the maximum value (see fig. 6). This optimal seed is subsequently refined with $n$-step denoising to produce the final image. Our approach quickly identifies good noise early, while VLM selection ensures the output aligns with human preferences. Further details are provided in the supplementary materials (Sup. Mat.).

Table 1: **Quantitative results on Emu Test set (§4.1).** Following [4, 3], we compute CLIP-I and DINO scores between the source and edited image, while CLIP-out measures the distance between output caption and edited image. We also employ GPT-4o to evaluate the edited results. The Train. Pa. means parameters finetuned for the editing task. * indicates methods that rely on output captions.

| Methods | Base Model | Train. Pa. | Data Usage | CLIP-I ↑ | CLIP-Out ↑ | DINO ↑ | GPT ↑ |
|---|---|---|---|---|---|---|---|
| InstructP2P [CVPR23] | SD 1.5 | 0.9B | 0.45M | 0.856 | 0.292 | 0.773 | 0.36 |
| MagicBrush [NeurIPS23] | SD 1.5 | 0.9B | 0.47M | 0.877 | 0.298 | 0.807 | 0.48 |
| EmuEdit [CVPR24] | Close Source | 2.8B | 10M | 0.877 | 0.306 | 0.844 | **0.72** |
| UltraEdit [NeurIPS24] | SD 3 | 2.5B | 3M | 0.880 | 0.304 | 0.847 | 0.54 |
| FluxEdit [huggingface] | Flux.1 dev | 12B | 1.2M | 0.852 | 0.282 | 0.760 | 0.22 |
| FLUX.1 Fill [huggingface] | Flux.1 Fill | - | - | 0.794 | 0.273 | 0.659 | 0.24 |
| RF-Solver Edit* [ICML25] | Flux.1 dev | - | - | 0.797 | **0.309** | 0.683 | 0.32 |
| ACE++ [arXiv25] | Flux.1 Fill | 12B | 54M | 0.791 | 0.280 | 0.687 | 0.24 |
| ICEdit (ours) | Flux.1 Fill | 0.2B | 0.05M | **0.907** | 0.305 | **0.866** | 0.68 |

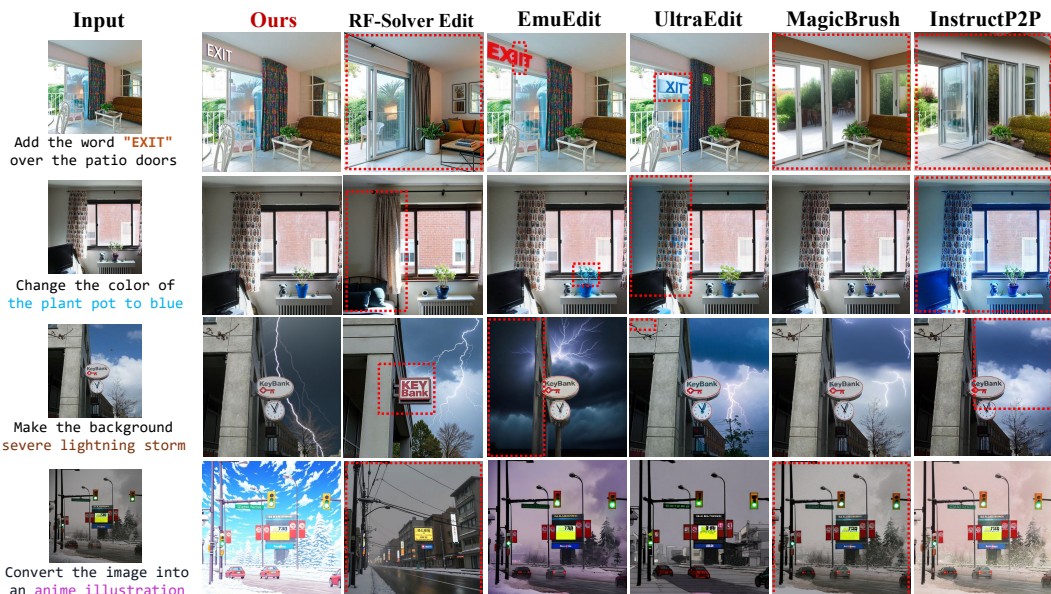

Figure 7: **Comparison with baseline models on the Emu Edit test set (§4.1).** Our method demonstrates superior performance in both edit-instruction accuracy and preservation of non-edited regions compared to the baseline models. 🔍 **Zoom in** for detailed view.

## 4 Experiment

**Implementation Details.** We use FLUX.1 Fill, a leading open-source DiT-based inpainting model, as our backbone. For fine-tuning our hybrid LoRA-MoE module, we curated a 50K-sample editing dataset, combining 9K samples from MagicBrush [2] and 40K from OmniEdit [34] to address MagicBrush's limitations in edit type balance, style-focused data, and domain diversity. The model employs a LoRA rank of 32, four MoE experts, and a TopK value of 1. For inference-time scaling, we use Qwen-VL-72B [51] to evaluate image outputs.

**Evaluation Settings.** We evaluated our model on Emu [3] and MagicBrush [2] test sets. For MagicBrush, which provides ground truth (GT) edited images, we follow [4, 2] to compute CLIP [54, 55], DINO [56, 57], and L1 metrics, measuring

Table 2: **Quantitative results on MagicBrush test set.** Following [4], all metrics are calculated between the edited image and GT edited image provided by MagicBrush [2].

| Methods | L1 ↓ | CLIP-I ↑ | DINO ↑ |
|---|---|---|---|
| InstructP2P | 0.114 | 0.851 | 0.744 |
| MagicBrush | 0.074 | 0.908 | 0.847 |
| UltraEdit | 0.066 | 0.904 | 0.852 |
| FluxEdit | 0.114 | 0.779 | 0.663 |
| FLUX.1 Fill | 0.192 | 0.795 | 0.669 |
| RF-Solver Edit* | 0.112 | 0.766 | 0.675 |
| ACE++ | 0.195 | 0.741 | 0.591 |
| ICEdit (ours) | **0.060** | **0.928** | **0.853** |

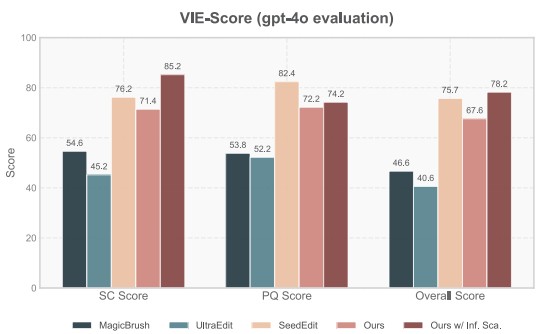 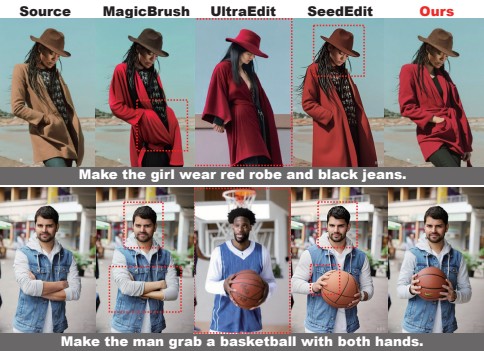

Figure 8: We employ the VIE-score to evaluate human preference alignment and quantify improvements from our inference-time scaling strategy (w/ Inf. Sca.) (§4.1 and §4.2).

Table 3: Ablation study on model structure configuration and inference time scaling settings.

(a) **Ablation study on model structure (§4.2).**

| Settings | Params | CLIP-I ↑ | CLIP-T ↑ | GPT ↑ |
|---|---|---|---|---|
| Training-free w/o IC prompt | - | 0.681 | 0.258 | 0.14 |
| Training-free w/ IC prompt | - | 0.794 | 0.273 | 0.24 |
| Only MoE module | 130M | **0.929** | 0.300 | 0.51 |
| LoRA (r=64) w/ IC prompt | 240M | 0.911 | 0.301 | 0.60 |
| Ours w/o IC prompt | 214M | 0.896 | 0.300 | 0.62 |
| Ours | 214M | 0.907 | **0.305** | **0.68** |

(b) **Ablation of inference time scaling (§4.2).**

| Verifier | Noise Num | Inf. Step | NFE ↓ | GPT ↑ |
|---|---|---|---|---|
| - | 1 | - | 50 | 0.68 |
| VLM | 6 | 10 | 110 | 0.78 |
| VLM | 6 | 4 | 74 | 0.72 |
| VLM | 12 | 10 | 170 | 0.79 |
| VLM | 6 | 50 | 350 | 0.80 |
| CLIP | 6 | 50 | 350 | 0.65 |

divergence from GT. For the Emu test set, lacking GT, we adopt baseline assessments from [4, 3] and use GPT-4o [58], same as [34], to assess editing success (see Sup. Mat.). All models use a single default noise input, **excluding our Early Filter Inference Time Scaling for fair comparison**.

Conventional metrics like CLIP [54, 55] and DINO [56, 57] often misalign with human preferences [59, 34, 10]. We thus employ VIE-Score [59], combining SC (instruction adherence and unedited region preservation) and PQ (visual quality) scores, computed as Overall $= \sqrt{\text{SC} \times \text{PQ}}$. This metric evaluates the improvement brought about by our inference-time scaling strategy and benchmarks our model against SeedEdit [60], a leading close-source model.

## 4.1 Comparisons with State-of-the-Art

**Results on Emu Edit and MagicBrush Test Sets.** We compare our model against UNet-based [1, 2, 3] and DiT-based [4, 42, 13, 7] methods (tables 1 and 2). Our model achieves SOTA-comparable performance, with MagicBrush outputs (table 2) closely matching GT and showing strong editing proficiency. On Emu (table 1), it aligns well with instructions while preserving image fidelity better than SOTA. GPT-based scores surpass open-source models and rival closed-source Emu Edit, despite using 0.5% training data. Compared to DiT-based models, our approach excels with fewer samples and parameters, demonstrating high efficiency. Qualitative results are in fig. 7 and Sup. Mat.

**VIE-Score Eval.** As shown in fig. 8, our model significantly outperforms open-source SOTA methods in editing accuracy and visual quality. Random seed testing shows performance nearing SeedEdit, and with our inference scaling strategy, it surpasses SeedEdit in overall VIE-Score. Although SeedEdit achieves higher PQ scores due to its polished outputs, it struggles with identity preservation in unedited regions. Our method, however, excels in maintaining fidelity in these areas (fig. 8).

## 4.2 Ablation Study

**Model Structure.** We validate our approach through experiments (table 3). The in-context edit prompt (IC prompt) significantly outperforms direct instructions in training-free models (70% GPT score increase) and enhances editing after fine-tuning. Our LoRA-MoE design outperforms standard LoRA, boosting quality and success rates (13% GPT score increase) with fewer parameters. Limiting adaptation to the output projection layer ("Only MoE") reduces performance, highlighting the need for comprehensive fine-tuning across all modules.

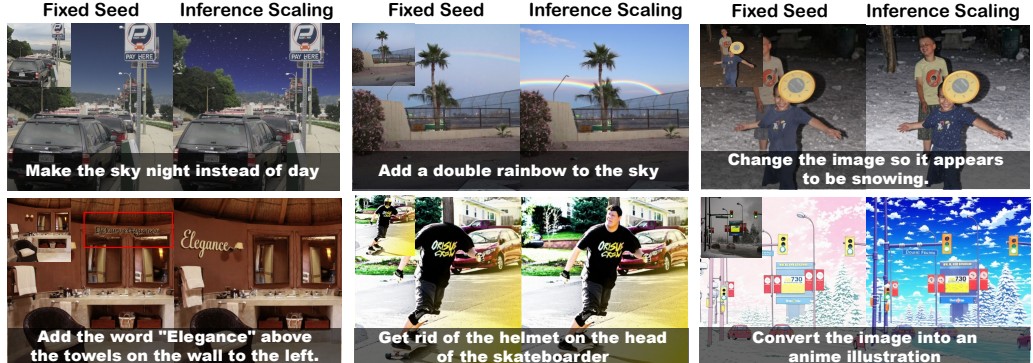

Figure 9: **Ablation on Inference-Time Scaling (§4.2).** Our strategy significantly enhances edit quality. For example, with the instruction "get rid of the helmet," default fixed seed incorrectly removes the character's head—a flawed outcome prevented by VLM filtering.

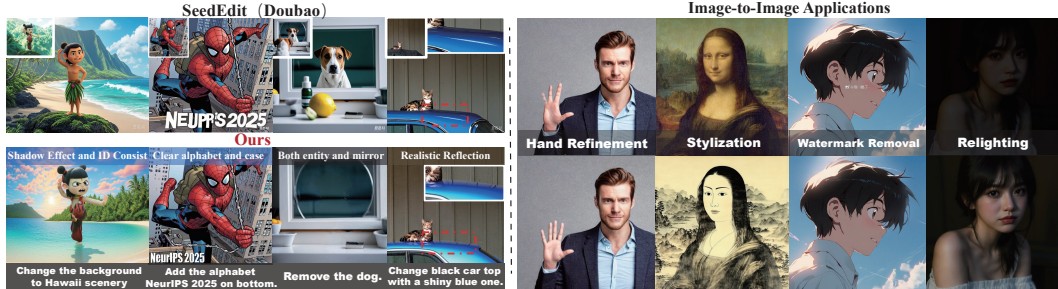

Figure 10: **Applications (§4.3).** Our method leverages DiT's original generation ability, producing harmonious results. Without additional tuning, it shows robust generalization across diverse tasks.

**Inference-Time Scaling.** As shown in figs. 8 and 9, our strategy markedly improves editing performance, achieving a 19% increase in SC score and a 16% boost in overall VIE-Score. Quantitative experiments across various settings (table 3) demonstrate that our approach significantly reduces computational cost (number of function evaluation, NFE) while substantially enhancing performance.

**Data Efficiency.** As shown in fig. 2 and table 1, our method yields significant improvements with only 0.05M training samples, achieving a 180% GPT-score increase over our training-free framework while using just 0.1% of samples required by prior models. This highlights the efficiency and effectiveness of our fine-tuning strategy.

## 4.3 Application

**Harmonious and Versatile Editing.** Our method leverages the powerful generative prior of large-scale Diffusion Transformers (DiTs) to create seamless, context-aware edits that blend naturally with the original image, automatically incorporating shadow effects and style alignment as shown in figs. 1 and 10. As a versatile image-to-image framework, it excels in tasks such as hand refinement and relighting (fig. 10) and holds potential for broader applications through task-specific fine-tuning.

## 5 Conclusion

In this paper, we present ICEdit, a novel DiT-based instruction editing method that delivers state-of-the-art performance with minimal fine-tuning data, achieving an unmatched balance of efficiency and precision. We first explore the inherent editing potential of generative DiTs in a training-free context, proposing our in-context edit paradigm. We then enhance its editing quality and robustness through minimal fine-tuning with the mixture of expert structure. Additionally, we introduce an early filter inference-time scaling strategy, using VLMs to select optimal early-stage outputs from multiple seeds, enhancing edit outcomes. Extensive experiments confirm our method's effectiveness and showcase superior results. We believe this efficient, precise framework establishes a new paradigm for balancing precision and efficiency in instructional image editing

**Acknowledgements.** This work was supported by the National Natural Science Foundation of China (62441617) and the Fundamental Research Funds for the Central Universities (226-2025-00080). This work was also supported by the Earth System Big Data Platform of the School of Earth Sciences, Zhejiang University.

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

# A  Preliminary

**Diffusion Transformer (DiT) Model** [20], employed in architectures such as FLUX.1 [23], Stable Diffusion 3 [61], and PixArt [21], utilizes a transformer as a denoising network to iteratively refine noisy image tokens. A DiT model processes two types of tokens: text condition tokens $C_\mathrm{T} \in \mathbb{R}^{M \times d}$ and noisy image tokens $X \in \mathbb{R}^{N \times d}$, where $d$ is the embedding dimension, and $M$ and $N$ represent the number of text and image tokens, respectively. These tokens maintain consistent shapes as they pass through multiple transformer blocks within the network.

**FLUX and FLUX-Fill** are based on a hybrid architecture that combines multimodal and parallel diffusion transformer blocks, scaled to 12B parameters. FLUX-Fill serves as both an inpainting and outpainting model, enabling modification of real and generated images using a text description and binary mask. Both models are built on flow matching, a general and conceptually simple method for training generative models, with diffusion as a special case.

In these models, each DiT block consists of layer normalization followed by Multi-Modal Attention (MMA) [62], which incorporates Rotary Position Embedding (RoPE) [63] to encode spatial information. The multi-modal attention mechanism projects the position-encoded tokens into query $Q$, key $K$, and value $V$ representations. This enables the computation of attention across all tokens:

$$\mathrm{MMA}([C_\mathrm{T}; X]) = \mathrm{softmax}\left(\frac{QK^\top}{\sqrt{d}}\right) V, \tag{3}$$

where $[C_\mathrm{T}; X]$ denotes the concatenation of text and image tokens, facilitating bidirectional attention.

# B  Implementation Details

## B.1  Dataset

As outlined in the main text, our fine-tuning dataset comprises 50K samples, a volume substantially smaller than the training data utilized by state-of-the-art (SOTA) models. The dataset is exclusively derived from open-access resources: MagicBrush [2] (9K samples) and OmniEdit [34] (40K randomly selected samples). The distribution of task types within our dataset is presented in Table table 4. It is worth noting that the dataset was not rigorously curated, and as illustrated in fig. 11, it still contains a number of problematic samples. Due to the time-intensive nature of data cleaning, **we opted not to perform additional filtering at this stage**. However, we anticipate that future efforts involving meticulous curation and the incorporation of higher-quality datasets could further enhance model performance.

Importantly, our model demonstrates superior performance compared to those trained on MagicBrush [2] and the full OmniEdit 1.2M dataset [42], despite utilizing significantly less data. This underscores the effectiveness of our in-context edit methodology, **suggesting that its advantages extend beyond the dataset itself**.

Table 4: Dataset Statistics by Task Type

| Task Type | Removal | Addition | Swap | Attribute Mod. | Style | Total |
|-----------|---------|----------|------|----------------|-------|-------|
| **Count** | 13,272 | 11,938 | 5,823 | 11,484 | 10,530 | 53,047 |

## B.2  Training-free Framework

Here we elaborate on the implementation of the in-context edit framework based on the T2I DiT. Our core idea is to guide the T2I DiT to generate side-by-side images, where the left side reconstructs

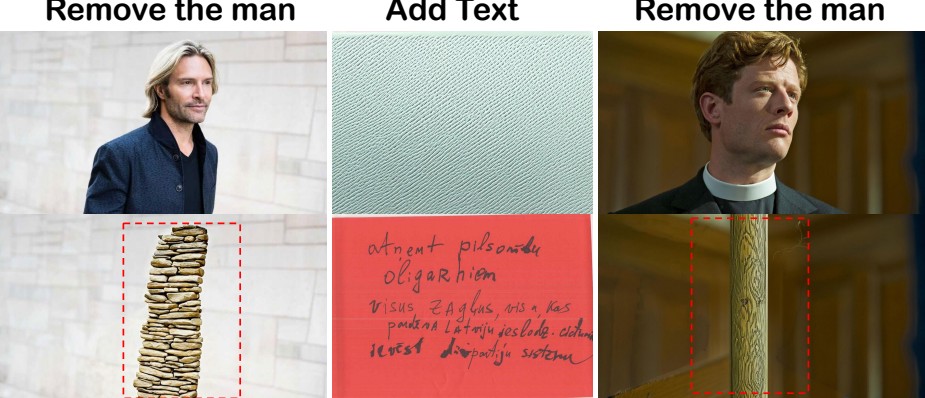

Figure 11: **Presence of Some Low-Quality Samples in the Dataset**. Our examination reveals that the public dataset contains a number of suboptimal samples, which could potentially impact the performance of our method.

the reference image, enabling the right side to incorporate features from the left image and perform reference based generation.

Specifically, we first perform image inversion using the T2I model [13, 14], retaining the value features from the attention layers during the inversion process. Subsequently, we generate images using the in-context edit prompt. To achieve this, we concatenate the noise obtained from the inversion process on the left side with a random noise of the same size on the right side as the input noise. It is crucial to apply positional encoding to the concatenated random noise to distinguish it from the inversion noise.

During the denoising steps, we inject the retained value features from the inversion process into the inversion noise portion, while leaving the random noise portion unaltered. This ensures that the left side of the image reconstructs the source image, while the right side, by leveraging the injected value features through the attention mechanism, generates a result with the same identity but conforming to the editing instructions. fig. 12 exhibits some results of this framework, which shows great potential for instruction-based image editing, despite minor artifacts in identity preservation and layout maintenance.

### B.3 Finetuning and Inference

Our model employs a Mixture of Experts (MoE) module with 4 expert LoRAs, each of rank 32. The routing network consists of a single linear layer with TopK set to 1, which balances increased parameter capacity with computational efficiency. For other modules, we use standard LoRA with the same rank of 32, and all LoRA alpha values are set equal to their respective ranks.

The model is trained with a batch size of 1 and gradient accumulation over 2 steps, resulting in an effective batch size of 2. We utilize the Prodigy optimizer [64] with safeguard warmup and bias correction enabled, configured with a weight decay of 0.01. Training is conducted on 4 A800 (80G) GPUs for one day, while inference is performed on an A100 (40G) GPU. Following [27], we optimize the parameters by minimizing the reconstruction loss between the model's predictions and the ground truth. Notably, we do not incorporate an additional load-balancing loss for the routing network, as our experiments reveal that the expert usage predicted by the router is well-balanced, with no single expert being overused. However, this approach may not hold for more complex MoE architectures, and we plan to explore this further in future work.

During training, input images are resized to **512×512** pixels, then formatted into **512×1024** diptychs (side-by-side image pairs). Here we report the detailed VRAM consumption under various settings.

**Memory Usage with Gradient Checkpointing:**

- **512×512 resolution**: 37 GB VRAM (batch size = 1)
- **768×768 resolution**: 39 GB VRAM (batch size = 1)

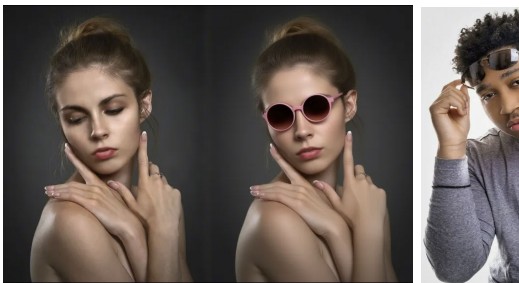 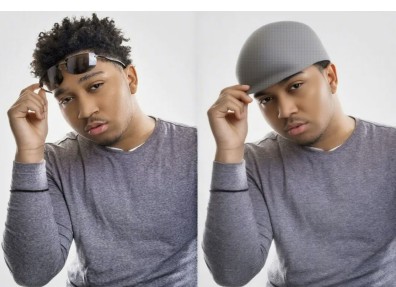



Make the girl wear pink sunglasses     Change the sunglasses to a white headless duck tongue cap



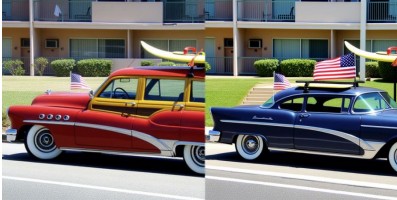 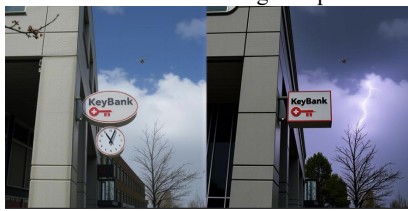



Alter the car color from red to blue     Make the backgroundsevere lightning storm



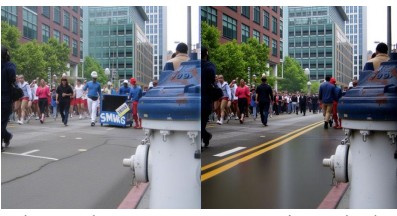 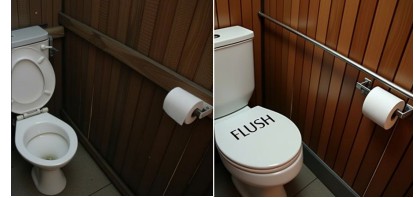



Change the street to appear rain-soaked.     Add the word "FLUSH" to the toilet seat.



Figure 12: **Results of training-free framework.** Our training-free framework demonstrates significant potential for instruction-based image editing, despite minor artifacts in identity preservation and layout maintenance.

- **1024×1024 resolution**: 42 GB VRAM (batch size = 1)

**Memory Usage without Gradient Checkpointing:**

- **512×512 resolution**: 60GB VRAM (batch size = 1)

- **768×768 resolution**: 77 GB VRAM (batch size = 1)

- **1024×1024 resolution**: Out of memory (OOM)

Given that gradient checkpointing significantly slows training speed and the baseline models are optimized for 512 resolution, we train our model at 512 resolution without gradient checkpointing to balance computational efficiency and memory constraints.

For both training and inference, we use the in-context (IC) prompt: "A diptych with two side-by-side images of the same scene. On the right, the scene is exactly the same as on the left but {instruction}." This allows the model to directly utilize editing instructions without additional adjustments. During inference, we set the guidance scale to 50 and the number of inference steps to 50. When employing the early filter inference time scaling strategy, we randomly select 6 noise samples for fast inference with 10 steps, then use Qwen2.5 VL 72B (via API) to pairwise evaluate the images and select the one that best aligns with the editing instruction for the next round of filtering. The prompt used for filtering is shown in·fig. 13. We also conduct experiments on parameter settings for inference time scaling, as detailed in appendix D.

"You are a multimodal large-language model tasked with evaluating images generated by a text-to-image model. You are given three images:
 1. The edited image1.
 2. The edited image2.
 3. The original image.

The edit_image_1 and edit_image_2 are generated by the edit prompt '{*instruction*}' and the original image. Please carefully compare the two edited images with the original image and answer the question: which of the two edited images (edit_image_1 and edit_image_2) better aligns with the edit prompt '{instruction}' and has higher quality and less artifacts.

Moreover, the areas that are not intended for editing should remain identical to the original image and the style should be consistent with the original image unless the edit prompt specifies otherwise.

Answer with 1 or 2 and briefly explain why."

Figure 13: **Prompt used for VLMs during inference time scaling.**

## B.4 Evaluation Details

### B.4.1 Baseline Details

In the main paper, we compare our method with both traditional and state-of-the-art models on the instruction-based image editing task. Below, we provide a brief overview of each baseline method:

- **InstructPix2Pix** [1]: This method fine-tunes Stable Diffusion [65] using automatically generated instruction-based image editing data. It enables instruction-based image editing during inference without requiring any test-time tuning.
- **MagicBrush** [2]: MagicBrush curates a well-structured editing dataset with detailed human annotations and fine-tunes its model using the InstructPix2Pix [1] framework. This approach emphasizes high-quality data for improved editing performance.
- **Emu Edit** [3]: Emu Edit is a closed-source model trained on a large-scale dataset of 10M samples for image editing. It introduces the Emu test dataset to evaluate editing quality. Since only the results on the Emu test set are publicly available, we compare our method with Emu Edit exclusively on this dataset.
- **Flux Edit** [42]: This is an open-source model on Hugging Face, fine-tuned on the Flux.1 dev model using 1.2M editing pairs from OmniEdit [34]. We include this baseline to highlight that our improvements stem primarily from our proposed framework rather than the base model (FLUX).
- **FLUX Fill** [23]: It is a 12 billion parameter rectified flow transformer capable of filling areas in existing images based on a text description. We use it as our training-free baseline for comparison with our framework.
- **RF-Solver-Edit** [13]: This is a training-free image editing framework based on FLUX, which requires image inversion followed by feature injection for editing. Since it cannot directly utilize instruction prompts, we provide the input and output captions from the dataset for generation. Due to the use of output captions, its CLIP-out metric tends to be higher.
- **ACE++** [7]: ACE++ is an enhanced version of ACE [6], trained on FLUX with a richer dataset of 700M samples, including 54M editing pairs. It integrates reference image generation, local editing, and controllable generation into a single framework. However, our tests reveal that it underperforms on instruction-based tasks.

- **SeedEdit** [60]: A state-of-the-art commercial model (based on DiT) trained on a meticulously curated large-scale dataset. This model achieves an optimal balance between *image reconstruction* (preserving original content) and *image re-generation* (creating novel visual content). Due to its limited accessibility through a web-based interface without API support, we conducted evaluations using a randomly curated subset of 50 images from the EMU test dataset and in-the-wild samples. Each image was processed manually through SeedEdit's interactive web interface to ensure consistent and reliable results.

### B.4.2 Metrics

MagicBrush is designed to evaluate the single-turn and multi-turn image editing capabilities of models. It provides annotator-defined instructions, editing masks, and ground truth images. Following the setup of [2, 4], we use the L1 metric to measure pixel-level differences between the generated image and the ground truth. Additionally, we employ CLIP similarity and DINO similarity to evaluate the overall resemblance to the ground truth.

The Emu Edit Test addresses bias and overfitting in annotator-defined datasets by eliminating ground truth images, while enhancing diversity through creative and challenging instructions paired with high-quality captions that capture essential elements in both source and target images. Consistent with [4], we evaluate the model's ability to preserve source image elements using CLIP image similarity and DINO similarity between **source images** and **edited images**. For text-image alignment, we employ CLIP-Out to measure the correspondence between local descriptions and generated image embeddings. Following [2, 4], we use ViT-B/32 for CLIP and dino_vits16 for DINO embeddings.

Notably, we exclude CLIP text-image direction similarity due to its inconsistency in reflecting editing quality, as it frequently assigns low scores to well-edited results (see fig. 14). Instead, we incorporate GPT-4o for complementary scoring. Furthermore, to mitigate benchmark quality issues—such as placeholder captions (e.g., 'a train station in city') or identical source-target captions—we filter out incorrect cases before evaluation, following [4]. While the Emu Edit Test successfully reduces image-level bias by omitting ground truth images, the evaluation metrics still implicitly assess the model's editing capability through feature-level comparisons.

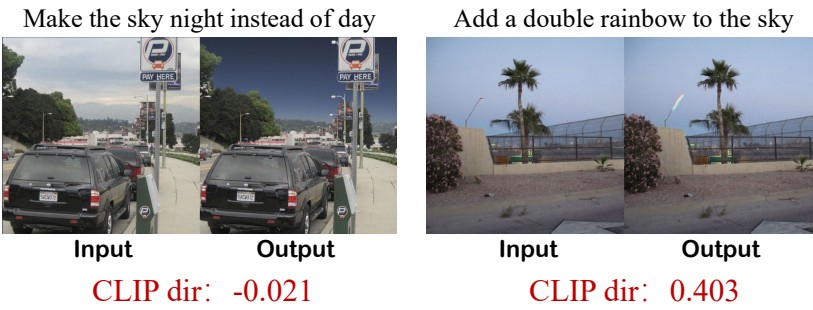

Figure 14: Our analysis reveals limitations in using CLIP direction scores for editing performance evaluation. Notably, the first example demonstrates a successful edit despite receiving a low score, while the second case shows a failed edit with an anomalously high score, highlighting the metric's inconsistency.

**GPT Evaluation**. Following [59, 34, 10], we employ GPT-4o to compute the VIE-score for assessing editing performance, which better aligns with human perceptual judgment. The evaluation prompts are detailed in figs. 15 and 16, where the *SC score* quantifies instruction adherence and editing accuracy, while the *PQ score* measures perceptual quality and naturalness. Consistent with [34], we apply a threshold-based binarization to the SC score, mapping it to the [0,1] range as a weighting function for the final GPT evaluation score.

## C   Discussion

**Limitations.** Despite achieving state-of-the-art editing performance with efficient tuning, our method suffers from the following limitations (fig. 17): (1) *Object Movement*: Instructions requiring spatial

"
You are a professional digital artist tasked with evaluating the effectiveness of AI-generated images based on the given rules. Provide your output as follows (keep reasoning concise and short):
{ "score": [<score1>, <score2>],
"reasoning": "..."
}.
Do not output anything else. Two images will be provided: the first is the original AI-generated image, and the second is an edited version of the first. Your objective is to evaluate how successfully the editing instruction has been executed in the second image. Note that the two images might sometimes look identical due to editing failure. Use a scale from 0 to 10 for two scores:

1. Editing Success Score:
0: The edited image does not follow the editing instruction at all.
10: The edited image follows the instruction perfectly.
If the object in the instruction is absent in the original image, score is 0.

2. Degree of Overediting Score:
0: The edited image is completely different from the original.
10: The edited image is recognizably a minimally edited yet effective version of the original.
Output the scores in a list: "score": [<editing_success>, <overediting>], where 'score1' evaluates editing success and 'score2' evaluates the degree of overediting.
"

Figure 15: **Prompt used for evaluating SC score.**

"
You are a professional digital artist tasked with evaluating the effectiveness of an AI-generated image. All images and humans depicted are AI-generated, so privacy or confidentiality is not a concern. Focus solely on technical quality and artifacts, ignoring whether the context appears natural. Evaluate based on:

Distortions
Unusual body parts or proportions
Unnatural object shapes
Rate the image on a scale from 0 to 10:
0: Significant AI-artifacts present.
10: Artifact-free image.
Provide your output as: { "score": <integer>, "reasoning": "..." }, keeping reasoning concise and short. Do not output anything else.
"

Figure 16: **Prompt used for evaluating PQ score.**

relocation (e.g., "move the chair to the corner") may fail due to insufficient exposure to motion-oriented data in general editing datasets. Specialized datasets like [66] could address this through targeted fine-tuning.(2) *Semantic Understanding Limitations*: While T5 demonstrates strong text encoding capabilities, its semantic understanding remains constrained, particularly in resolving polysemous terms (e.g., confusing "mouse" (computer device) with "mouse" (animal)). This limitation stems from its limited contextual disambiguation ability. Future work could incorporate MLLM-based modules [8, 67] to improve semantic fidelity. (3) *VLM Efficiency*: Our inference time scaling relies on Qwen-VL 72B to ensure accurate quality assessment, as smaller models (7B) often misjudge edit quality. Recent advances [34, 68] demonstrate that distilled 7B VLMs can achieve GPT-4o-level performance through specialized training, offering a promising path toward efficiency improvements.

**Broader Impact.** Our work may contribute to the field in two aspects: (1) *Accessible Image Editing Tools*: The parameter-efficient design could reduce computational resource requirements, potentially benefiting small-scale developers and individual creators. (2) *Ethical Considerations*: Like most generative models, our technology could potentially be misused for creating deceptive content

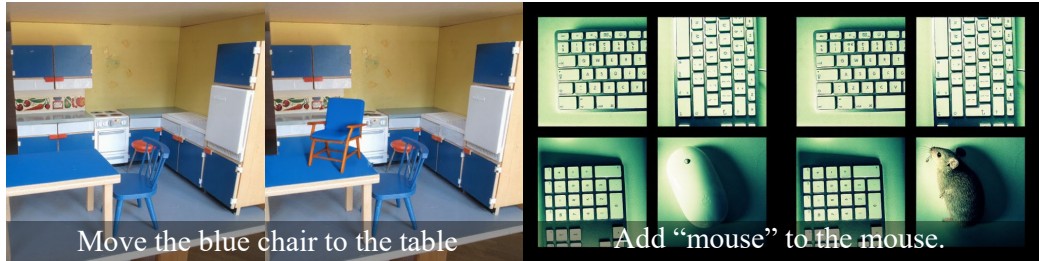

Figure 17: Some failure cases of our methods, such as object movement, semantic ambiguity.

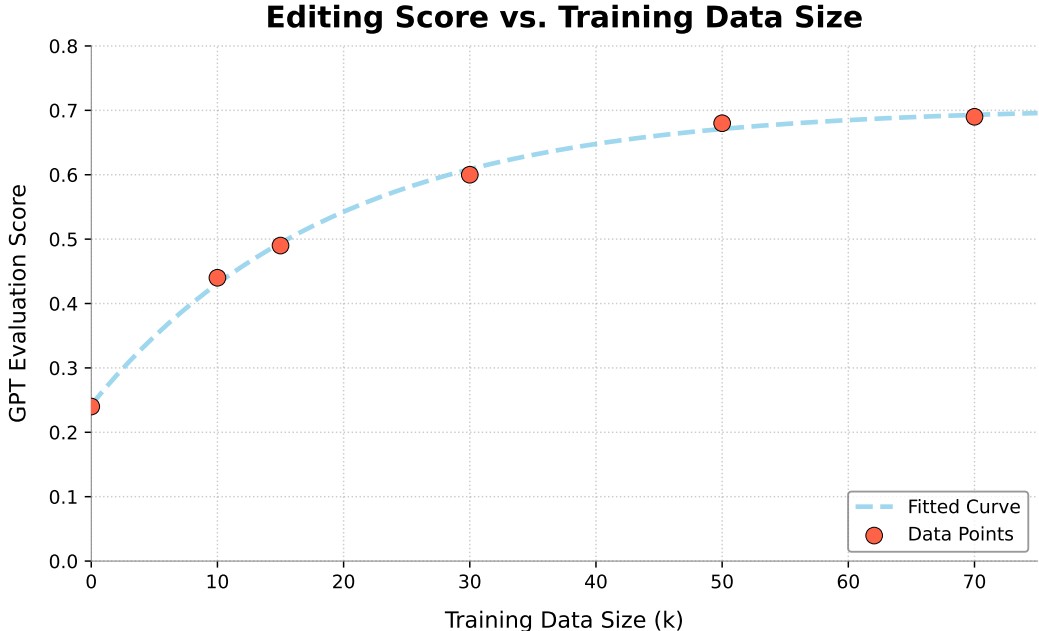

Figure 18: The plot demonstrates that the model's performance improves with increasing training data, with the growth rate gradually plateauing.

(e.g., deepfakes or manipulated media). We advocate three essential safeguards for responsible deployment: (1) implementing content provenance standards to ensure traceability, (2) maintaining human oversight in sensitive applications such as news media and legal documentation, and (3) fostering collaboration between AI developers and domain experts to establish ethical guidelines specific to instruction-based editing scenarios. These measures align with emerging AI governance frameworks while respecting creative freedoms.

## D    Additional Results

### D.1    Ablation

**MoE Settings.** We conduct a series of experiments to investigate the influence of different expert configurations. As shown in table 5, we observe a significant improvement in model performance (measured by GPT-based evaluation scores) when increasing the expert rank from 8 to 32 or the number of experts from 1 to 4. However, further increasing the number of experts does not lead to notable gains in editing performance, while significantly increasing the model's parameter count, thereby reducing efficiency. We hypothesize that as the number of experts grows, the routing mechanism becomes more challenging, potentially requiring more sophisticated routing network designs and corresponding loss constraints to achieve stable performance. Future research could explore these aspects to optimize the trade-off between performance and efficiency.

Table 5: **Ablation of different settings.**

| Expert Number | Expert Rank | Params | CLIP-I ↑ | CLIP-Out ↑ | GPT ↑ |
|:---:|:---:|:---:|:---:|:---:|:---:|
| 1 | 32 | 120M | 0.892 | 0.300 | 0.59 |
| 4 | 8 | 120M | 0.920 | 0.303 | 0.58 |
| 4 | 32 | 214M | 0.907 | 0.305 | 0.68 |
| 6 | 32 | 270M | 0.914 | 0.305 | 0.66 |
| 8 | 32 | 335M | 0.907 | 0.304 | 0.61 |

**Inference Scaling Settings.** We explore the impact of different parameter configurations on our early filter inference time scaling strategy, as detailed in table.3(b) in the main paper. Specifically, we investigate the choice of evaluator (VLM vs. CLIP), the number of random noise seeds, and the selection of early inference steps (early steps). The computational efficiency and accuracy are measured using NFE (Number of Function Evaluations, the cumulative compute count) and GPT-based evaluation scores. The total inference steps are fixed at 50, and NFE is calculated as $50 + (Num\_noise \times Early\_step)$.

Our experiments reveal that using 10 early steps outperforms 4 steps, as DiT may fail to generate sufficiently high-quality results with only 4 steps, leading to inaccurate VLM judgments. Increasing the number of initial noise seeds improves performance significantly from 1 to 6, but the gains diminish from 6 to 12. We attribute this to the model's robust editing performance across most noise samples, resulting in greater improvements compared to using a single noise seed.

Additionally, we adopt the strategy from [29], which uses traditional metrics like CLIP to filter results by selecting the image-text pair with the highest CLIP score after full-step inference. However, this approach increases computational cost while degrading GPT-based evaluation scores, indicating that CLIP-based scoring does not align well with human visual preferences.

**Data Efficiency.** We investigated how editing performance scales with training data volume, as shown in fig. 18. Fine-tuning with just 10K samples markedly improves performance over training-free approaches. As data size increases, editing capability continues to rise, though the rate of improvement gradually diminishes. Notably, training with 70K samples yields minimal gains over 50K, suggesting convergence under this setup. Future work could explore the effects of larger-scale parameter configurations and more high-quality editing data.

## D.2 Analysis on Expert Selection Patterns

**Implicit Expert Routing Across Layers and Steps.** In our current design, the MoE module does not explicitly assign a specific expert based on task type or instruction semantics. Instead, the MoE is integrated into each layer of the DiT backbone (e.g., 57 layers in Flux.1 Fill), and expert selection is performed independently at every layer, diffusion step, and token. As described in the main paper, each token at each layer and timestep is dynamically routed to one of the experts. For instance, generating a single image with 28 diffusion steps results in $28 \times 57 = 1{,}596$ expert selection events per token. These selections are distributed across all available experts and vary throughout the generation process, rather than consistently favoring a particular expert for a given task. Therefore, expert usage is inherently implicit and fine-grained, making direct attribution to instruction types non-trivial.

**Preliminary Analysis of Expert Usage Patterns.** To further explore this aspect, we conducted a targeted analysis of expert usage across different editing task types. Specifically, we selected three representative instruction categories from the `EmuEdit` test set—*Addition*, *Removal*, and *Style Editing*—and ran 50 inference samples for each category. We then recorded the expert selection frequencies across all layers, diffusion steps, and tokens (38,133,76 selections per inference). The distribution of expert usage is summarized in Table 6.

## D.3 Top-K Expert Selection.

We choose the expert TopK value to be 1, as it offers a strong trade-off between computational efficiency and predictive performance. Activating only a single expert per token significantly reduces

Table 6: Distribution of expert selection frequencies across editing task types. The MoE module demonstrates balanced expert utilization without strong bias toward specific tasks.

| Task Type | Expert 0 | Expert 1 | Expert 2 | Expert 3 |
|-----------|----------|----------|----------|----------|
| Addition | 25.4% | 25.4% | 24.1% | 25.0% |
| Removal | 25.9% | 26.2% | 23.2% | 24.7% |
| Style Editing | 24.7% | 24.6% | 25.0% | 25.7% |

FLOPs, memory usage, and communication overhead—critical factors for both training and real-time inference in deployment settings. This design choice is consistent with prior work: for instance, Switch Transformers [69], LLaVA-MoLE [70], and Mixtral [46] collectively demonstrate that Top-1 routing enables scalable sparse MoE training with minimal accuracy loss, while higher TopK values may lead to diminishing returns and higher computational costs.

We also experimented with TopK = 2 and TopK = 4 using four total experts. The results on the `EmuEdit` test set are summarized in Table 7.

Table 7: Performance comparison under different TopK values. Selecting more experts slightly degrades performance while increasing computational cost.

| TopK Value | CLIP-I ↑ | CLIP-OUT ↑ |
|------------|----------|------------|
| TopK = 1 (Ours) | 0.907 | 0.305 |
| TopK = 2 | 0.891 | 0.301 |
| TopK = 4 | 0.866 | 0.298 |

As shown above, increasing the number of selected experts leads to a slight but consistent drop in performance. Therefore, we adopt TopK = 1 as the optimal setting. We hypothesize that the lack of improvement with larger $K$ may stem from the absence of additional constraints guiding expert selection. Incorporating auxiliary losses or regularization to encourage more effective expert routing represents a promising direction for future work.

## D.4 Qualitative Results

Here we present additional qualitative results, including comparisons with baseline models (figs. 19 and 20), editing results with different initial noise configurations (fig. 21), and a broader range of editing examples (fig. 22). These results further illustrate the effectiveness and versatility of our approach across diverse scenarios.

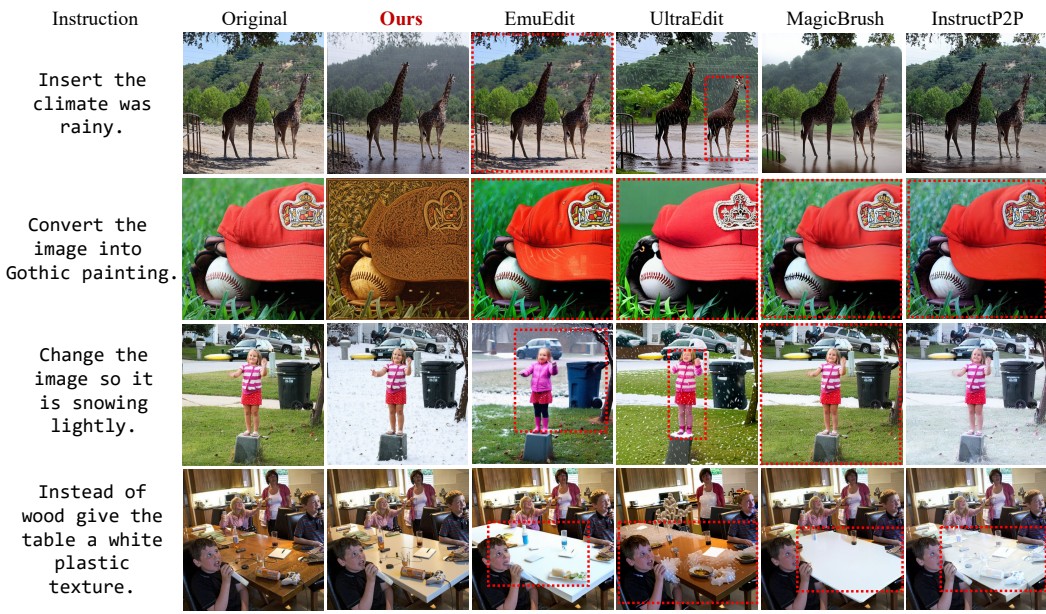

Figure 19: **Comparison with baseline models on Emu Edit test set.** 🔍 **Zoom in** for detailed view.

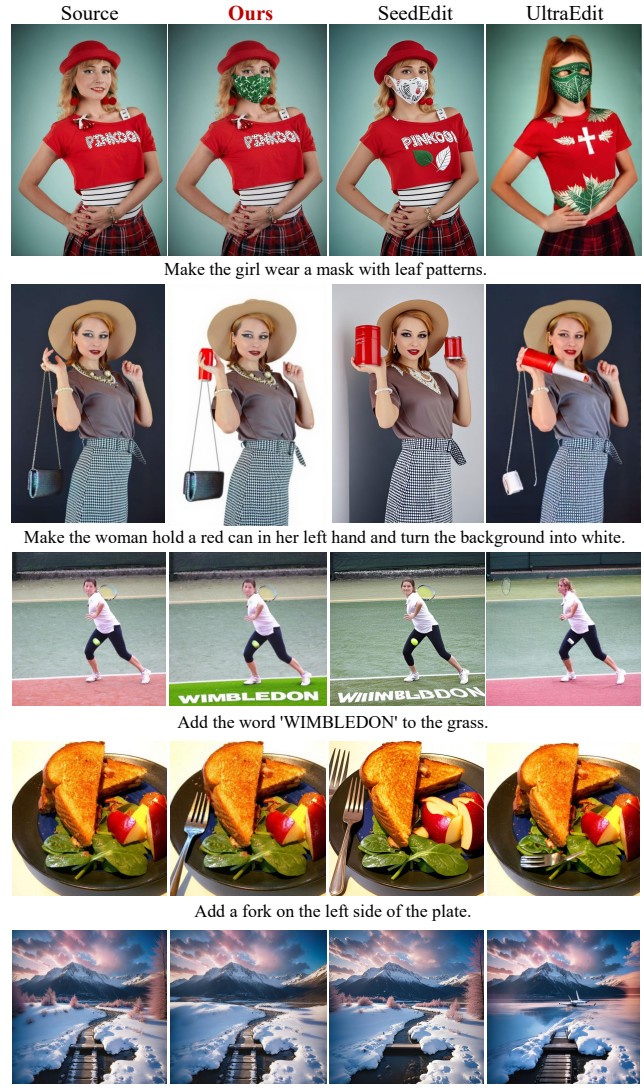

Figure 20: **Comparison with baseline models.** 🔍 **Zoom in** for detailed view.

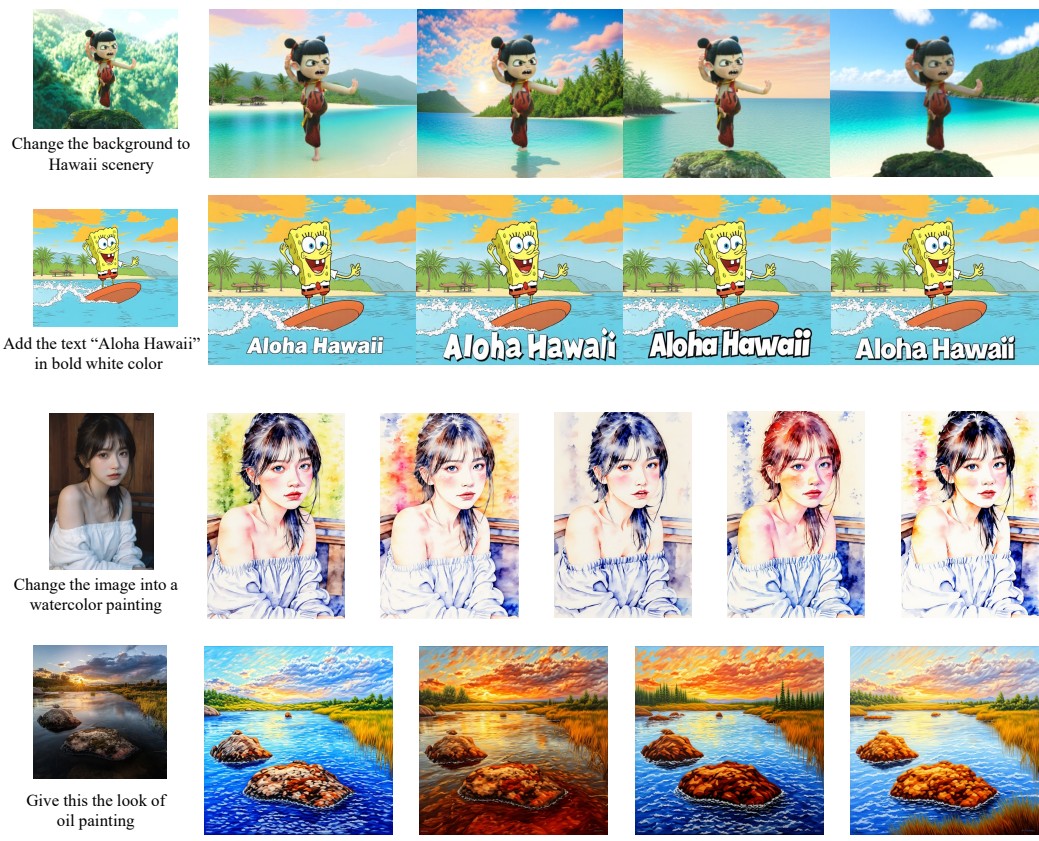

Figure 21: With different initial noise, our methods generate diverse results.

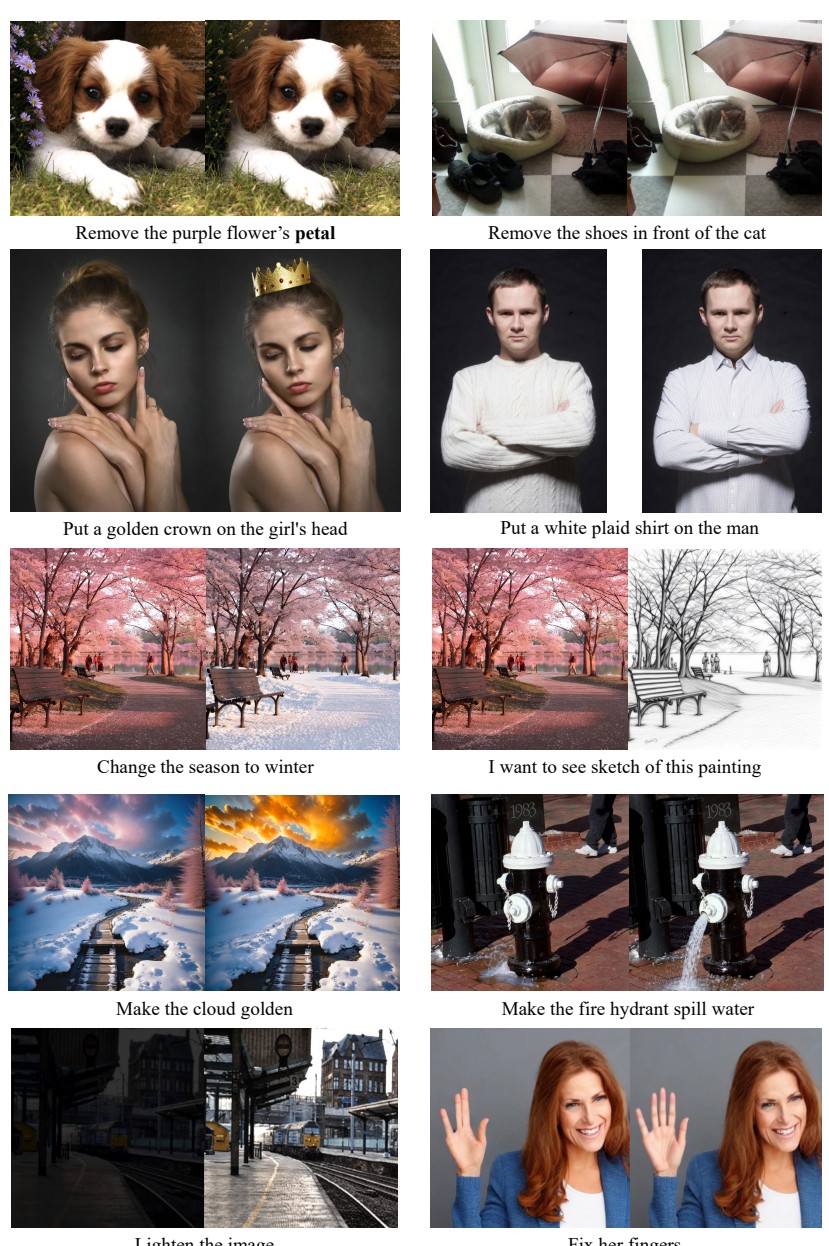

Figure 22: More editing results of our method.

