# OpenReview forum: "Enabling Instructional Image Editing with In-Context Generation in Large Scale Diffusion Transformer"
_NeurIPS.cc/2025/Conference — NeurIPS 2025 poster_

### Official Review · Reviewer_ZGQZ · 2025-06-29

**Clarity:** 3
**Significance:** 3
**Originality:** 3
**Rating:** 5
**Confidence:** 3

**Summary:**

The paper proposes ICEdit for the task of image editing. ICEdit minimizes the alteration of existing image diffusion transformer architecture, and requires very low fine-tuning efforts, but can still achieve similar and even better quality compared to baseline methods. ICEdit also utilizes an early filter inference-time scaling approach to select high-quality noise samples for generation efficiency and quality. ICEdit shows promising results on standard image editing benchmarks and with qualitative results.

**Questions:**

1. Does the proposed pipeline need to fine-tune a VLM judge? In the paper, it indicates to use a QwenVL-72B, may I ask the authors if any fine-tuning about image aesthetics is required for better judging?
2. In the visual illustration, I see ICEdit support object removal. May I ask if the task of object location manipulation (e.g., move the apple from left to right in the image) is also supported?
3. Though requiring much less computational resources to finetune compared to training-based baselines, utilizing VLM (especially a 72B model) and inference with multiple seeds may pose a computational difficulty. Would the author mind clarifying the computational cost during inference?

**Ethical Concerns:**

["NO or VERY MINOR ethics concerns only"]

**Final Justification:**

I am now fully convinced by the method’s pipeline, especially with the authors’ clear explanation regarding the need for user-provided location information or manual masks during inference, which I found quite surprising. Additionally, the use of VLM API calling appears to be an efficient and user-friendly approach, particularly since the huge VLM model does not require any fine-tuning. While object movement remains a challenge, I understand that this is a common limitation in image editing tasks. Overall, I am satisfied with the rebuttal, and most of my concerns have been fully addressed. Therefore, I would like to increase my score and recommend acceptance.

**Limitations:**

Yes, the authors adequately addressed the limitations and potential negative societal impact of their work.

**Quality:**

3

**Strengths And Weaknesses:**

Strength:
1. The proposed design choice on image editing prompting is carefully studied.
2. The proposed pipeline requires minimal fine-tuning effort and architectural changes, but results are improved compared to the baseline method.
3. The proposed test-time scaling with the early-stage intervention is very interesting and quite novel to image generation.

Limitations:
1. Even though the design choice of utilizing an inpainting-based approach is carefully studied and explained, this can be a limitation if users would like to have end-to-end editing with a prompt without any instruction on locations.

---

> ### Author Rebuttal · Authors · 2025-07-26
>
> Thank you sincerely for recognizing our design choices and acknowledging that our method is both novel and contributes to improved results. We truly appreciate your encouraging feedback.
>
> We address your questions and concerns point by point in the following responses.
>
> - **Answer to Limitation 1:** **Clarification Regarding Location Instruction and Inpainting Design**
>     1. Thank you for your comment. We would like to clarify a possible misunderstanding: **our method does not require any user-provided location information or manual masks**.
>         - Although we adopt an inpainting-based architecture, **the mask used during training and inference is a fixed binary mask of the same resolution as the input image**, always applied to the half of the diptych (**as stated in the main paper lines 18–19, 149–152** and **illustrated in Figure 5**). This design choice is internal to the model and does **not** require any user interaction or localization input.
>         - From the user’s perspective, **our system operates** **fully end-to-end**: **the user provides only an image and a natural language instruction**—**no specification of edit regions is needed**. **Our model is capable of automatically identifying and editing the appropriate regions based on instruction semantics**. Therefore, the model remains fully compatible with prompt-only editing workflows and does not pose limitations in terms of usability or interactivity.
>         - We appreciate the opportunity to clarify this point and will make it more explicit in future revisions to avoid similar misunderstandings.
>
> - **Answer to Question 1:  Clarification on the Use of VLM Judge and Fine-tuning**
>     1. Thank you for your insightful question. **In our current pipeline, we directly use the pre-trained Qwen-VL 72B model without any additional fine-tuning.** We chose this model because of its strong zero-shot capabilities, which we found sufficient for reliable judgment of image-editing quality and alignment.
>     2. We also experimented with smaller pre-trained variants such as Qwen-VL 7B and 32B. However, we observed that their judgment ability was less robust and less consistent across diverse editing cases, which led us to prefer the 72B version despite its larger scale.
>     3. That said, we agree that fine-tuning a smaller vision-language model (VLM) on editing-specific or aesthetics-related data is a promising alternative. Recent works such as **SANA 1.5**, **OmniEdit**, and **ImgEdit** have demonstrated that lightweight, fine-tuned VLMs can serve effectively as judges when trained on targeted datasets.
>     4. Our primary goal in this work was to explore the *inference-time scaling strategy* for editing models. Leveraging a powerful model like Qwen-VL 72B helped us validate the core effectiveness of our algorithm. In future work, we plan to explore replacing it with smaller, fine-tuned models to reduce computational costs while maintaining evaluation reliability.
>
> - **Answer to Question 2: Object Movement Editing Ability**
>     1. As discussed in the supplementary material (lines 151–155) and illustrated in Supplementary Figure 7, one known limitation of our model is its handling of **object movement**. Specifically, instructions involving spatial relocation (e.g., "move the chair to the corner") may lead to suboptimal results. **This is likely due to the limited exposure to motion-oriented transformations in standard editing datasets.** We believe this limitation could be mitigated through targeted fine-tuning on specialized datasets, such as the one introduced in *Cao, Mingdeng, et al., "Instruction-based Image Manipulation by Watching How Things Move."*
>     2. Addressing object movement remains an important direction for future work, and we plan to explore the integration of such specialized datasets to enhance our model’s spatial reasoning and relocation capabilities.
>     3. **That said, this limitation does not undermine the overall contribution of our work.** Compared to prior methods, our approach provides a **more efficient** and **higher-precision** framework for high-quality image editing, and consistently outperforms existing baselines across multiple benchmarks.
>
> - **Answer to Question 3: Clarification of Computational Cost During Inference**
>     1. Thank you for raising this important point. We agree that directly running a 72B VLM model can be computationally intensive. **However, in our implementation, we access the Qwen-VL 72B model via the official API, which eliminates any local GPU memory overhead (as mentioned in Sup. Mat. lines 82-84).** In practice, each API call (i.e., a single instruction-image pair evaluation) takes approximately **2 second**, making it efficient and practical in most real-world scenarios.
>     2. As for our editing model, the base inference setup on an A100 GPU consumes approximately **33 GB of memory and requires around 10 seconds to generate one edited image**. That said, with **CPU offloading** and lightweight quantization strategies such as **SVDQuant** [Li, Muyang, et al., *"Svdquant: Absorbing Outliers by Low-Rank Components for 4-bit Diffusion Models,"* arXiv:2411.05007], we can reduce the memory requirement to **under 10 GB**, significantly lowering the barrier to deployment.
>
>         These optimizations make our approach much more accessible while still maintaining strong performance.
>
> We hope this response addresses your concern, and we’re happy to further clarify if there are any remaining questions.
>
> **References:**
>
> - Xie, Enze, et al. "Sana 1.5: Efficient scaling of training-time and inference-time compute in linear diffusion transformer." *arXiv preprint arXiv:2501.18427* (2025).
> - Wei, Cong, et al. "Omniedit: Building image editing generalist models through specialist supervision." *The Thirteenth International Conference on Learning Representations*. 2024.
> - Ye, Yang, et al. "Imgedit: A unified image editing dataset and benchmark." *arXiv preprint arXiv:2505.20275* (2025).
> - Cao, Mingdeng, et al. "Instruction-based image manipulation by watching how things move." *Proceedings of the Computer Vision and Pattern Recognition Conference*. 2025.
> - Li, Muyang, et al. "Svdquant: Absorbing outliers by low-rank components for 4-bit diffusion models." arXiv preprint arXiv:2411.05007 (2024).

---

> > ### Comment · Reviewer_ZGQZ · 2025-08-04
> >
> > Thank you for the detailed response in the rebuttal.  I am now fully convinced by the method’s pipeline, especially with the authors’ clear explanation regarding the need for user-provided location information or manual masks during inference, which I found quite surprising. Additionally, the use of VLM API calling appears to be an efficient and user-friendly approach, particularly since the huge VLM model does not require any fine-tuning. While object movement remains a challenge, I understand that this is a common limitation in image editing tasks. Overall, I am satisfied with the rebuttal, and most of my concerns have been fully addressed. Therefore, I would like to increase my score and recommend acceptance.

---

> > > ### Author Response · Authors · 2025-08-04
> > >
> > > Thank you very much for your thoughtful feedback and for taking the time to carefully read our rebuttal. We sincerely appreciate your recognition of the proposed method and are glad that our clarifications regarding the pipeline, the use of VLMs, and the "mask-free design" helped address your concerns. We are also grateful for your understanding of the remaining challenges such as object movement. Your support and constructive comments have been truly encouraging, and we deeply appreciate your updated score and recommendation for acceptance.

---

### Official Review · Reviewer_NHpf · 2025-06-30

**Clarity:** 2
**Significance:** 3
**Originality:** 2
**Rating:** 4
**Confidence:** 5

**Summary:**

This paper proposes an in-context learning approach for instructional image editing, introducing two frameworks: one built on a text-to-image (T2I) DiT model and another based on an inpainting DiT model. The latter incorporates a parameter-efficient fine-tuning technique to further boost performance. Experimental results show that the method achieves state-of-the-art performance with minimal training data.

**Questions:**

My main concerns are outlined in the weaknesses section. To summarize, I would like the authors to address the following points in the rebuttal:

- What is the motivation for presenting the T2I DiT framework, given that it is not used in the experiments? This raises questions about the clarity and focus of the contributions.
- Important details about the dataset are missing, and clarification is needed to fully understand the experimental setup.

**Ethical Concerns:**

["NO or VERY MINOR ethics concerns only"]

**Final Justification:**

I appreciate the authors' response. Since they proposed renaming the method in the experiment to address my confusion, I am willing to raise my rating.

**Limitations:**

Yes, in supplemental materials. I would suggest to move it to the main paper.

**Quality:**

3

**Strengths And Weaknesses:**

Strengths

1. The idea of leveraging minimal, parameter-efficient learning is interesting.
2. The experimental results appear promising and demonstrate strong performance.

Weaknesses

1. The paper’s presentation is unclear, making it difficult to follow and obscuring the main contributions. It seems that the Mixture-of-Experts (MoE) fine-tuning is a key contribution. However, the authors present two methods: one with fine-tuning and one without. This dual-approach is confusing—if fine-tuning is a central contribution, the motivation for including the text-to-image DiT method (which does not involve fine-tuning and requires computationally expensive image inversion, as mentioned in Line 146) is unclear.

2. Assuming the main experiments are based on the fine-tuned inpainting DiT model—which takes a mask as input—it is not explained how these masks are generated in the training dataset.

3. Since the training data appears to be carefully curated, it raises concerns about potential data overfitting. Would the model still perform well if the training data were randomly sampled from standard datasets? A comparison with randomly sampled training data would help clarify this.

---

> ### Author Rebuttal · Authors · 2025-07-26
>
> Thank you for your positive feedback and for recognizing the value of our approach and strong performance of our method. We appreciate that very much!
>
> We greatly appreciate your feedback and will address your comments and concerns point by point in the following responses.
>
> - **Answer to Weakness 1 and Question 1: Clarification of Our Motivation and Methods**
>
>     We believe there may have been **a misunderstanding or oversight** regarding the clarity of our motivation and method.
>
>     Other reviewers have provided positive comments on this aspect, noting that “**the writing and illustrations are clear and well-organized, making the proposed method easy to understand,**” and commending the **“extensive analysis”** and **“clearly provided implementation details”** (**Reviewer oDrm**); that the **“method design is interesting and appears to be novel”** (**Reviewers oDrm and ZGQZ**); and that **“the introduction, technical explanation, and experimental sections are **all well-structured, with clear and coherent writing throughout**”** (**Reviewer oCfi**).
>
>     We will further clarify our motivation and method in the response below, with references to the relevant sections and figures in the paper to aid your understanding.
>
>     **Explanation:**
>
>     1. **Clarification of motivation:**
>         - Our primary goal is to leverage the generative capacity of large-scale DiT models for instruction-based image editing *without architectural modifications or heavy retraining* (as noted in Lines 19–26, 35–39).
>         - To this end, we propose a novel **in-context editing paradigm** (Fig. 2a), where the DiT model is reinterpreted as a "painter" that produces edited images (right panel) by understanding the reference image (left panel) and the instruction.
>         - **To explore the feasibility of this paradigm, we first introduce two training-free methods.** Although they demonstrate initial instruction-following capabilities, they suffer from **poor instruction comprehension** and **layout instability** (as shown in Figs. 3 and 4, and discussed in Lines 40–43).
>         - To address these issues, we enhance the editing capability under our paradigm (the inpainting-based paradigm) by integrating **IC Prompt**, **parameter-efficient fine-tuning via MoE-LoRA with minimal data**, and a **test-time scaling strategy** (Lines 44–60).
>         - Remarkably, this approach achieves **state-of-the-art performance** using only **0.1% of the training data** compared to prior models.
>     2. **Clarification of two training-free approaches:**
>         - **The two training-free approaches share the same core purpose** — to validate our proposed **in-context editing paradigm** (Fig. 2a). Whether based on a T2I DiT or an inpainting DiT architecture, both aim to generate an edited result (right panel) conditioned on the reference image (left panel) and instruction (Lines 110–119).
>         - **These variants serve as an important proof of concept**: despite the absence of fine-tuning, both models demonstrated some ability to perform in-context editing, indicating that **DiT models inherently possess the potential to support this new editing paradigm**.
>         - However, **we also found that the performance of these training-free variants is limited**. As shown in Fig. 4, the outputs are often unstable or of low quality. Quantitatively, both variants achieved similarly low GPT scores (**0.24**), as reported under “Training-free w/ IC prompt” in **Table 3(a)**—far below the results of our fine-tuned models.
>         - Therefore, the claim that *“the T2I DiT framework is not used in the experiments”* is inaccurate—it was explicitly used as one of our training-free baselines to validate the feasibility of our paradigm.
>         - Motivated by the observed limitations of the training-free variants, we proceeded to fine-tune the inpainting-based DiT model, which is more computationally efficient than its T2I counterpart. The inpainting DiT directly supports denoising-based editing, avoiding the need for inversion and reducing runtime by roughly half (Lines 140–147).
>
>         In summary, the training-free experiments establish two key findings （Lines 140–147）:
>
>         (1)**DiT models are capable of in-context editing**
>
>         (2)**Fine-tuning is necessary to fully realize the potential and stability of the proposed paradigm**.
>
>     3. **The finetuned model:**
>         - Building on the inpainting DiT, we improve editing capability through a combination of parameter-efficient MoE-LoRA fine-tuning on a small dataset and a test-time scaling strategy (Sections 3.2, 3.3, and 4), ultimately achieving SOTA performance.
>         - **Our contributions span from proposing the motivation and validating the paradigm with training-free methods, to designing a lightweight fine-tuning strategy and inference-time enhancement for stable, high-quality editing** （Lines 69-78, Lines 243-251）.
>         - **Focusing solely on the fine-tuning or on the T2I architecture overlooks the broader scope of our work** — we are introducing a **novel and general in-context editing framework** that unifies learning-free and learning-based components under a single paradigm.
>
> - **Answer to Weakness 2: Use of Mask in the Model**
>     1. **Our model does not require any manually created local masks.** **As stated in the main paper (lines 18–19, 149–152)** and **illustrated in Figure 5**, we simply use a fixed-size binary mask that matches the resolution of the input image. This mask is always applied to the right part of the diptych, while the left part provides the unmasked reference image.
>     2. Although we utilize an inpainting model (as our final solution), **our system functions like any other instruction-based image editing method**: the user only needs to provide an instruction and an image—**no explicit indication of the edit region is needed**. **Our model is capable of automatically identifying and editing the appropriate regions based on instruction semantics**. We believe this misunderstanding may have led to confusion about our method’s design. We appreciate your question and will clarify this point more explicitly in future revisions to avoid such misinterpretations.
>
> - **Answer to Weakness 3 and Question 2: Details About the Training Dataset**
>     1. **Our training data was not carefully curated but rather randomly sampled from standard datasets.** As stated in the supplementary material **(lines 20–28)**, *“the dataset was not rigorously curated,”* and it indeed contains many **problematic samples**, as illustrated in Supplementary Figure 1. We did **not** perform any additional filtering or cleaning of the dataset. Therefore, the assumption that our dataset was carefully selected is incorrect.
>   2. Regarding the OmniEdit data we use, we provide details on our download and usage process here. Since the OmniEdit dataset contains over 500 parquet files with a variety of edit types, we sampled a subset of files across different types to maintain diversity. **All parquet files were directly released by the dataset authors and randomly generated**—**we did not apply any post-selection or filtering**. Specifically, we selected 17 out of the 571 files: 1, 25, 50, 100, 115, 130, 160, 175, 225, 250, 300, 350, 425, 525, 566, 568, and 570. These files cover various editing types, including addition, replacement, removal, and attribute modification. The last few files (566–570) correspond to style editing, which constitutes only a small portion of the full dataset.
>   3. Importantly, **the use of randomly sampled data did not negatively impact model performance**. As shown in Supplementary Figure 8, increasing the amount of randomly sampled training data leads to consistent improvements, demonstrating a clear **scaling effect**. **This suggests that our model generalizes well without overfitting to any curated subset.**
>   4. Due to space limitations, **the discussion of dataset details was included in the supplementary material (Section B)**. We apologize if this caused you to overlook this information, and we will make these details more prominent in the main paper in future revisions. Besides, we will release the full codebase and all training configurations to facilitate reproducibility. The exact data sources, file IDs, and download instructions for the subset of OmniEdit used in our experiments will be clearly documented in the open-sourced repository.
>
> We hope the explanation above resolves your concern. Please feel free to let us know if further clarification is needed.

---

### Official Review · Reviewer_oCfi · 2025-07-02

**Clarity:** 3
**Significance:** 3
**Originality:** 3
**Rating:** 4
**Confidence:** 4

**Summary:**

This paper proposes ICEdit, an instruction-based image editing framework built upon a large-scale Diffusion Transformer (DiT). By leveraging an in-context prompt structure, a lightweight LoRA-MoE fine-tuning mechanism, and a noise filtering strategy during inference (Early Filter Inference-Time Scaling), ICEdit achieves high-quality editing performance under extremely limited training data and parameter constraints. This work advocates for minimal intervention and architectural modification to activate the native contextual understanding and generative capabilities of DiT, thereby balancing the trade-off between editing precision and inference efficiency observed in traditional methods.

**Questions:**

Please refer to weaknesses

**Ethical Concerns:**

["NO or VERY MINOR ethics concerns only"]

**Final Justification:**

The author's response has addressed most of my concerns, including clarifications on distinctions from prior methods, discussions on the limitations of the adopted approach, and additional analysis regarding parameter selection. Therefore, I am inclined to raise the original score. I appreciate the author's efforts during the rebuttal process.

**Limitations:**

yes

**Quality:**

3

**Strengths And Weaknesses:**

Strength:
1.	The paper proposes ICEdit, which leverages the original Diffusion Transformer (DiT) model to perform instruction-based image editing through an in-context approach—offering a novel and practical direction.
2.	It introduces the Early Filter Inference-Time Scaling strategy, which utilizes a vision-language model (VLM) to filter early-stage outputs, mitigating instability caused by initial noise.
3.	It also presents a LoRA-MoE structure to enhance the model’s capability in handling diverse tasks; this combination of LoRA and MoE is both meaningful and effective in practice.
4.	The introduction, technical explanation, and experimental sections are all well-structured, with clear and coherent writing throughout.

Weakness:
1.	Existing works such as Prompt-to-Prompt and Textual Inversion have explored similar directions, so it would be helpful to clarify how this approach differs from those prior attempts.
2.	The paper also appears to lack an analysis of the model’s limitations—for example, a discussion of failure cases or incorrect edits could provide a more balanced evaluation.
3.	While the overall experiments are thorough, the choice of TopK=1 in the MoE method is not sufficiently justified and would benefit from further explanation.

---

> ### Author Rebuttal · Authors · 2025-07-26
>
> Thank you sincerely for recognizing our method as *“**offering a novel and practical direction**”* and *“**both meaningful and effective in practice**,”* as well as for your positive comments on the overall structure and clarity of our writing. We truly appreciate your encouraging feedback.
>
> We’ll address your questions and concerns point by point below, and we strictly adhere to the double-blind review policy.
>
> - **Answer to Weakness 1: Difference from Previous Methods such as P2P and Textual Inversion**
>     1. **Our proposed training-free approaches are fundamentally different from prior methods such as P2P and Textual Inversion.**
>         - **Conceptual distinction**: **Our method is designed based on our proposed in-context editing paradigm (Fig. 2a)**, where the DiT model is reinterpreted as a "painter" that generates the edited image (right panel) by understanding both the reference image (left panel) and the instruction. This is conceptually distinct from previous training-free approaches like P2P or Textual Inversion, which do not operate under such a paradigm.
>         - **Difference from P2P**: Prompt-to-Prompt [Hertz et al., 2022] relies on descriptive prompts and cross-attention feature swapping to modify generation behavior. In contrast, our method performs instruction-guided editing, where the model is directly guided by the instruction to perform edits—**without requiring complex operations such as attention map replacement, addition, or deletion**.
>         - **Difference from Textual Inversion**: Textual Inversion [Gal et al., 2022] learns a new token to represent a visual concept from multiple reference images, and generates novel instances of that concept. This is a **different task altogether** from instruction-guided editing of a **single input image**, which Textual Inversion is not designed to handle.
>         - While we do adopt general-purpose techniques such as inversion and feature injection in our T2I-based variant — similar to P2P or Textual Inversion — **these are widely used in the field** (e.g., in MasaCtrl, PnP Inversion, RF-Solver, StableFlow) and serve only as tools to implement our paradigm rather than define it.
>         - Importantly, after exploring training-free realizations, we further refine the proposed framework via **few-shot fine-tuning with minimal data**, achieving **significant performance gains** that far surpass previous methods. This highlights that our proposed **in-context editing paradigm** is not only novel, but also demonstrably more effective than prior attempts.
> - **Answer to Weakness 2: Limitations of our model**
>     1. **We have included a discussion of the model’s limitations in the supplementary material (Section C, lines 151–162 and Fig. 7).** Due to space constraints, this section was not included in the main paper, but we will make its presence more prominent in a future revision.
>
>         Specifically, our model may produce incorrect edits or failure cases in the following aspects:
>
>         - **Object movement**: Instructions involving spatial relocation (e.g., "move the chair to the corner") may not be executed correctly. This is likely due to the lack of motion-oriented examples in general editing datasets. Targeted fine-tuning with specialized datasets could help address this issue.
>         - **Semantic understanding limitations**: Some failures arise from the model’s limited ability to resolve contextual ambiguities. Incorporating MLLM-based modules in future work could enhance semantic fidelity and instruction interpretation.
>
>         That said, these limitations do **not compromise the overall significance** of our results. Compared to prior work, our method offers a **more efficient** and **higher-precision** framework for high-quality image editing, and demonstrates consistent advantages across multiple benchmarks.
>
> - **Answer to Weakness 3:** **Choice of expert selection**
>
>     Thank you for recognizing the thoroughness of our experimental evaluation. Here are explanations of the expert selection.
>
>     1. **We choose expert TopK = 1 because it offers** **a strong trade-off between computational efficiency and predictive performance (as mentioned in lines 175-183 in Sup. Mat.)**. Activating only a single expert per token significantly reduces FLOPs, memory usage, and communication overhead—critical for both training and real-time inference in deployment settings. This design is well-supported by prior work: for instance, **Switch Transformers** [Fedus et al., 2022] and **LLaVA-MoLE** [Liu et al., 2024] demonstrate that Top-1 routing enables scalable sparse MoE training with minimal loss in accuracy. Furthermore, **Mixtral** [Beaumont et al., 2023] shows that while TopK > 1 may yield marginal gains in some cases, it comes at the cost of significantly increased computation and memory, which may not align with real-world constraints. Given these trade-offs and our deployment-oriented motivation, we adopt TopK = 1 as a practical and effective choice. We will add a clarification and references in the final version.
>     2. We have experimented with settings using TopK = 2 and TopK = 4 with a total of 4 experts. The results on the EmuEdit test set are summarized below:
>
>
>         | TopK Value | CLIP-I ↑ | CLIP-OUT ↑ |
>         | --- | --- | --- |
>         | **TopK = 1 (Ours)** | **0.907** | **0.305** |
>         | TopK = 2 | 0.891 | 0.301 |
>         | TopK = 4 | 0.866 | 0.298 |
>
>         **As shown above, increasing the number of selected experts leads to a slight but consistent drop in performance.** Therefore, we chose **TopK = 1** as the optimal setting. We will clarify this design choice more explicitly in a future version of the paper.
>         We hypothesize that the lack of improvement with larger K may be due to the absence of additional constraints to guide expert selection. Incorporating auxiliary losses or regularization to encourage more effective expert routing is a promising direction for future work.
>
>
> We hope the explanation above resolves your concern. Please feel free to let us know if further clarification is needed.
>
> References:
>
> - Hertz, Amir, et al. "Prompt-to-prompt image editing with cross attention control." *arXiv preprint arXiv:2208.01626* (2022).
> - Gal, Rinon, et al. "An image is worth one word: Personalizing text-to-image generation using textual inversion." *arXiv preprint arXiv:2208.01618* (2022).
> - Cao, Mingdeng, et al. "Masactrl: Tuning-free mutual self-attention control for consistent image synthesis and editing." *Proceedings of the IEEE/CVF international conference on computer vision*. 2023.
> - Ju, Xuan, et al. "Direct inversion: Boosting diffusion-based editing with 3 lines of code." *arXiv preprint arXiv:2310.01506* (2023).
> - Wang, Jiangshan, et al. "Taming rectified flow for inversion and editing." *arXiv preprint arXiv:2411.04746* (2024).
> - Avrahami, Omri, et al. "Stable flow: Vital layers for training-free image editing." *Proceedings of the Computer Vision and Pattern Recognition Conference*. 2025.
> - Fedus, William, Barret Zoph, and Noam Shazeer. "Switch transformers: Scaling to trillion parameter models with simple and efficient sparsity." *Journal of Machine Learning Research* 23.120 (2022): 1-39.
> - Chen, Shaoxiang, Zequn Jie, and Lin Ma. "Llava-mole: Sparse mixture of lora experts for mitigating data conflicts in instruction finetuning mllms." *arXiv preprint arXiv:2401.16160* (2024).
> - Jiang, Albert Q., et al. "Mixtral of experts." *arXiv preprint arXiv:2401.04088* (2024).

---

> > ### Comment · Reviewer_oCfi · 2025-08-05
> >
> > The author's response has addressed most of my concerns, including clarifications on distinctions from prior methods, discussions on the limitations of the adopted approach, and additional analysis regarding parameter selection. Therefore, I am inclined to raise the original score. I appreciate the author's efforts during the rebuttal process.

---

> > > ### Author Response · Authors · 2025-08-05
> > >
> > > **Dear Reviewer oCfi,**
> > >
> > > Thank you very much for your thoughtful response and for indicating that you are inclined to raise the original score — we truly appreciate it.
> > >
> > > We’re glad to hear that the rebuttal has addressed most of your concerns, including the clarifications on prior methods, limitations of our approach, and parameter analysis. If there are any remaining questions or concerns, we’d be more than happy to continue the discussion.
> > >
> > > We’re also sincerely grateful for your kind words in the original review, including your comments that **“ICEdit offers a novel and practical direction,”** **“the Early Filter Inference-Time Scaling strategy mitigates instability caused by initial noise,”** **“the combination of LoRA and MoE is both meaningful and effective in practice,”** and that **“the introduction, technical explanation, and experimental sections are all well-structured, with clear and coherent writing throughout.”** Your recognition is both encouraging and deeply appreciated.
> > >
> > > Best regards,
> > >
> > > *The Authors*

---

### Official Review · Reviewer_oDrm · 2025-07-06

**Clarity:** 3
**Significance:** 3
**Originality:** 3
**Rating:** 5
**Confidence:** 4

**Summary:**

This paper tackles the problem of instruction-based image editing, which aims to modify images via natural language prompts while balancing precision and efficiency. The authors propose ICEdit, a novel method that leverages the capabilities of Diffusion Transformers (DiTs) through (1) an in-context editing paradigm, (2) minimal parameter-efficient fine-tuning, and (3) an inference-time filtering strategy using Vision-Language Models. ICEdit achieves state-of-the-art results with only 0.1% of the training data and 1% of trainable parameters compared to existing approaches.

**Questions:**

While I believe the paper is of high quality overall, I encourage the authors to further validate the diptych-based editing framework and provide a deeper analysis of expert selection behavior in the LoRA-MoE module.

**Ethical Concerns:**

["NO or VERY MINOR ethics concerns only"]

**Final Justification:**

I carefully read the authors’ rebuttals and the reviews from other reviewers. The rebuttals effectively address my original concerns on the effectiveness of diptych-based designs and on the LoRA-MoE model. Thus, I will maintain my original rating of “accept.”

**Limitations:**

yes

**Paper Formatting Concerns:**

No concerns on paper formatting for this submission.

**Quality:**

4

**Strengths And Weaknesses:**

Strengths

1. The overall quality of the paper is high.
- The writing and illustrations are clear and well-organized, making the proposed method easy to understand. The paper also includes extensive analysis, such as an exploration of scaling effects with respect to training data — an aspect I was particularly curious about while reading. Furthermore, important implementation details, including prompts for VLM-based inference-time scaling and evaluation setups, are clearly provided.

2. The idea of diptych-based image editing and the corresponding method design are interesting and appear to be novel.

3. ICEdit demonstrates impressive performance, achieving a strong balance between editing quality and training efficiency. This claim is well-supported by comprehensive experimental results.

Weaknesses

1. While the diptych-based image editing formulation is compelling, its effectiveness is not fully validated.
- To better isolate and validate the contribution of the diptych-based design, it would be helpful to compare against a baseline that does not use the diptych structure but retains other components, such as the LoRA-MoE module and training dataset. In this setup, the baseline would generate an edited image solely based on the original image and instruction.

2. Lack of analysis on the expert selection patterns in the LoRA-MoE module.
- The paper claims that the LoRA-MoE design improves editing quality, and the final model appears to select one LoRA expert among four. It would be valuable to investigate whether there are consistent patterns in expert selection depending on factors such as task type or instruction category.

---

> ### Author Rebuttal · Authors · 2025-07-28
>
> Thank you sincerely for your positive and encouraging feedback, including remarks such as **“The overall quality of the paper is high,” “The writing and illustrations are clear and well-organized,” “Important implementation details are clearly provided,” “The idea and the corresponding method design are interesting and appear to be novel,” and “Demonstrates impressive performance.”**  We truly appreciate your thoughtful evaluation, and we’re glad to see that the key strengths of our work were recognized.
>
> We address your inquiries and concerns point by point in the following responses.
>
> - **Answer to Weakness 1 and question 1:** **Ablation Study on the Diptych-Based Design**
>     1. Thank you for your insightful and constructive suggestion. Following your advice, we conducted an ablation study by removing the diptych structure. Instead, we adopted a channel-wise concatenation of the reference image and noise (as in InstructPix2Pix) as input to the model. We retained the MoE-LoRA module and fine-tuned on Flux.1 Dev using the **same training dataset** and **the same number of GPUs**.
>
>         The performance comparison on the EmuEdit benchmark is summarized below:
>
>         | Model Variant | CLIP-I ↑ | CLIP-OUT ↑ | DINO ↑ |
>         | --- | --- | --- | --- |
>         | **ICEdit (Ours)** | 0.907 | 0.305 | 0.866 |
>         | w/o Diptych | 0.837 | 0.275 | 0.748 |
>         | FluxEdit (baseline) | 0.852 | 0.282 | 0.760 |
>
>         As shown above, **removing the diptych structure leads to a noticeable performance drop across all metrics, confirming the effectiveness of our proposed design.**
>
>     2. A similar observation can be found in one of the baseline models already included in our paper. As shown in Table 1 and Table 2, the **FluxEdit** baseline—which also uses a channel-wise concatenation of the reference image and noise, and is trained from scratch on the full 1.2M OmniEdit dataset—achieves relatively poor results, further supporting our findings.
>     3. We believe the performance degradation in the non-diptych setting stems from the fact that **channel-wise concatenation alters the input encoding pattern, forcing the model to adapt to a different image structure and latent space.** This often destabilizes the output unless the model undergoes heavy retraining, which is especially challenging for large-scale DiT backbones. We find this to be a very meaningful point and will include a discussion on it in the revised version of the paper.
>
> - **Answer to Weakness 2: Expert Selection Patterns in the LoRA-MoE Module**
>
>     Thank you for your thoughtful question. We agree that analyzing expert selection behavior is a valuable direction, and we address it in two parts below.
>
>     1. **Implicit Expert Routing Across Layers and Steps**
>
>         In our current design, the MoE module does not explicitly assign one expert based on task type or instruction semantics. Instead, the MoE is integrated into **each layer** of the DiT backbone (e.g., 57 layers in Flux.1 Fill), and **expert selection is performed independently at every layer, every diffusion step, and for each token**. As described in the main paper (lines 161–165), each token at each layer and timestep is routed to one of the experts dynamically. For example, generating a single image with 28 diffusion steps results in **28 × 57 = 1,596 expert selection events per token**. These selections are distributed across all available experts and vary throughout the generation process, rather than consistently favoring a particular expert for a given task. Therefore, expert usage is inherently **implicit and fine-grained**, making direct attribution to instruction types non-trivial.
>
>     2. **Preliminary Analysis of Expert Usage Patterns**
>
>         To further explore your suggestion, we conducted a targeted analysis of expert usage across different editing task types. Specifically, we selected three representative instruction categories from the EmuEdit test set—**Addition**, **Removal**, and **Style Editing**—and ran **50 inference samples** for each category. We then recorded the expert selection frequencies across all layers, diffusion steps, and tokens (38,133,76 selections per inference).
>
>         The distribution of expert usage is summarized below:
>
>         | Task Type | Expert 0 | Expert 1 | Expert 2 | Expert 3 |
>         | --- | --- | --- | --- | --- |
>         | Addition | 25.4% | 25.4% | 24.1% | 25.0% |
>         | Removal | 25.9% | 26.2% | 23.2% | 24.7% |
>         | Style Editing | 24.7% | 24.6% | 25.0% | 25.7% |
>
>         These results show that expert usage remains **relatively balanced across different instruction types**, indicating that **all experts are actively contributing within the model** rather than being underutilized. This also reinforces our earlier point that expert selection is inherently implicit and fine-grained, and does not directly align with high-level semantic categories such as task type.
>
>         We consider this an insightful direction and plan to conduct more in-depth analysis in future work to further understand expert dynamics and potential specialization patterns.

---

### Note · Authors · 2025-08-13

We sincerely thank all reviewers for their efforts during the review and discussion phases. We are encouraged that our key contributions were explicitly recognized, and we appreciate the constructive suggestions that will further improve our work.

**1. Novel in-context editing paradigm with large pretrained DiTs**

Our main idea—using a diptych-based *in-context* paradigm for instruction-based editing—was noted as *“novel and practical”* (**oCfi**) and *“interesting and… novel”* (**oDrm**). This enables editing without architectural changes or heavy fine-tuning, as **ZGQZ** observed, while outperforming baselines.

**2. Parameter-efficient enhancement: LoRA-MoE & Early Filter Inference-Time Scaling**

Our **LoRA-MoE** and **Early Filter** strategies were described as *“meaningful and effective”* (**oCfi**), *“very interesting and quite novel”* (**ZGQZ**), and as *“leveraging minimal, parameter-efficient learning”* (**NHpf**). They improve editing precision while keeping efficiency.

**3. Strong performance–efficiency balance**

**oDrm** highlighted ICEdit’s *“impressive performance”* and *“strong balance between quality and efficiency,”* supported by extensive experiments including scaling analyses. Our method reaches SOTA while using only **0.1% training data** and **1% trainable parameters**.

**4. Clarity and reproducibility**

We are grateful for comments such as **Reviewer oCfi**’s praise that *“the introduction, technical explanation, and experimental sections are all well-structured, with clear and coherent writing throughout”*, and **Reviewer oDrm**’s appreciation of *“clear and well-organized”* writing with *“important implementation details… clearly provided.”* We also value **Reviewer NHpf**’s and **Reviewer oDrm**’s constructive suggestions, which we will integrate to further improve readability and deepen analysis.

In summary, our contributions—novel in-context editing with large DiTs, parameter-efficient enhancement strategies, and strong performance–efficiency balance—were all explicitly acknowledged by the reviewers. We believe ICEdit offers a fresh perspective for instruction-based image editing and can inspire further research in this area.

---

### Decision · Program_Chairs · 2025-09-17

**Decision:**

Accept (poster)

**Comment:**

This paper introduces ICEdit, an instruction-based image-editing framework that contains three components: (1) an in-context “diptych” prompting scheme, (2) a lightweight fine-tuning module, and (3) an Early-Filter Inference-Time Scaling strategy. Extensive experiments across multiple benchmarks demonstrate that ICEdit achieves superior performance.

Overall, reviewers find the technical contribution solid and the experimental results convincing. But meanwhile, some concerns are raised: 1) the effectiveness of the diptych prompting scheme should be more carefully ablated; 2) insufficient analysis on the MoE expert behaviour; 3) more details are needed regarding the usage and computational overhead of the 72B VLM; 4) it is unclear how inpainting masks are generated; and 5) the presentation clarifty can be further enahnced.

The rebuttal is considered, which successfully addresses most of these concerns. As a result, all reviewers unanimously recommend acceptance. For the final version, please ensure that all promised revisions from the rebuttal are fully integrated to ensure the clarity and quality of this NeurIPS publication.